# Site representativity of AERONET and GAW remotely sensed AOT and AAOT observations

**Nick A.J. Schutgens**[1]

[1]Department of Earth Science, Vrije Universiteit Amsterdam, 1081 HV Amsterdam, the Netherlands

**Correspondence:** Nick Schutgens (n.a.j.schutgens@vu.nl)

**Abstract.** Remote sensing observations from the AERONET (AErosol RObotic NETwork) and GAW (Global Atmosphere Watch) networks are intermittent in time and have a limited field-of-view. A global high-resolution simulation (GEOS5 Nature Run) is used to conduct an Observing System Simulation Experiment (OSSE) for AERONET and GAW observations of AOT (Aerosol Optical Thickness) and AAOT (Absorbing Aerosol Optical Thickness) and estimate the spatiotemporal representativity of individual sites for larger areas (from $0.5^o$ to $4^o$ in size).

GEOS5 NR and the OSSE are evaluated and shown to have sufficient skill, although daily AAOT variability is significantly underestimated while the frequency of AAOT observations is over-estimated (both resulting in an underestimation of temporal representativity errors in AAOT).

Yearly representation errors are provided for a host of scenarios: varying grid-box size, temporal collocation protocols, and site altitudes are explored. Monthly representation errors show correlations from month to month, with a pronounced annual cycle that suggests temporal averaging may not be very successfull in reducing multi-year representation errors. The collocation protocol for AEROCOM (AEROsol Comparisons between Observations and Models) model evaluation (using daily data) is shown to be sub-optimal and the use of hourly data is advocated instead. A previous subjective ranking of site *spatial* representativity (Kinne et al., 2013) is analysed and a new objective ranking proposed. Several sites are shown to have yearly representation errors in excess of 40%.

Lastly, a recent suggestion (Wang et al., 2018) that AERONET observations of AAOT suffer a positive representation bias of 30% globally is analysed and evidence is provided that this bias is likely an overestimate (the current paper finds 4%) due methodological choices.

## 1 Introduction

As the temporal sampling of observations is often intermittent and their field-of-view limited, the ability of observations to represent the weather or climate system is negatively affected (Nappo et al., 1982). This adverse effect can be described through a representation error, which allows comparison to e.g. observational errors or model errors.

Representation errors have been receiving more attention recently, in a variety of fields: solar surface radiation (Hakuba et al., 2014b, a; Schwarz et al., 2017, 2018), sea surface temperatures (Bulgin et al., 2016), trace gases (Sofieva et al., 2014; Coldewey-Egbers et al., 2015; Lin et al., 2015; Boersma et al., 2016), water vapour (Diedrich et al., 2016), cloud susceptibility (Ma et al., 2018) and even climate data (Cavanaugh and Shen, 2015; Director and Bornn, 2015). In the field of aerosol, most work has been on the representativity of satellite measurements (Kaufman et al., 2000; Smirnov, 2002; Remer et al., 2006; Levy et al., 2009; Colarco et al., 2010; Sayer et al., 2010; Colarco et al., 2014; Geogdzhayev et al., 2014), either using satellite data or model data. A new development is the use of local spatially relatively dense measurement networks (Shi et al., 2018; Virtanen et al., 2018).

As aerosols are known to vary over short time and spatial scales (Anderson et al., 2003; Kovacs, 2006; Santese et al., 2007; Shinozuka and Redemann, 2011; Weigum et al., 2012; Schutgens et al., 2013), aerosol studies are likely to experience large representation errors. Indeed, Schutgens et al. (2016b) (S16b hereafter) showed that representation errors due to temporal sampling in both satellite and AERONET (AErosol RObotic NETwork) observations were of similar magnitude as actual model errors and often larger than observational errors. Similarly, Schutgens et al. (2016a) (S16a hereafter) showed that the narrow field-of-view of in-situ measurements could lead to large differences from area av-

erages (monthly RMS differences of $10-80\%$ for $210 \times 210$ km$^2$, depending on the type of measurement and the location of the site). Recently, Schutgens et al. (2017) (hereafter S17) considered the combined impact of spatio-temporal sampling on the representativeness of remote sensing data (both satellite and ground-based). They provide representation uncertainty estimates and optimal strategies when dealing with different observing systems (ground networks, polar orbiting satellites with varying revisit times, or geo-stationary satellites).

In this paper, a global one-year high-resolution simulation of the atmosphere (GEOS5 Nature Run) is used to conduct an Observing System Simulation Experiment to estimate representation errors for remote sensing measurements of aerosol optical thickness (and its absorptive counterpart) as observed by the global networks AERONET and GAW (Global Atmosphere Watch). In S16a and S17, regional high-resolution simulations covering a month were used to study representation errors. This prevented an analysis of such errors world-wide and on longer time-scales. In addition, the limited spatio-temporal domains made evaluation of the high-resolution simulation difficult. These issues are addressed in the current study. Note that the current paper does not replace previous work (which also considers satellite, in-situ and flight measurements) but extends it. In addition, the current study allows us to evaluate a recent suggestion by Wang et al. (2018) that representation errors in AERONET AAOT observations are positively biased (by $\sim 30\%$) which would help to explain the observed underestimation of AAOT in global models (Bond et al., 2013).

Representation errors are not only determined by observational sampling but also by how these observations are put to use. If observations are used to evaluate models, different protocols (or strategies) exist to temporally collocate model data and observations. For instance, within AEROCOM (AEROsol Comparisons between Observations and Models), an oft-used strategy is daily collocation: daily averages of observations are collocated with daily model data. The different sampling of model and observations throughout the day are ignored (e.g. most remote sensing observations only observe a small part of the diurnal cycle). In contrast, hourly collocation uses hourly model data that is collocated with hourly averages of observations. S17 showed that in the case of remote sensing observations daily collocation allows significantly larger representation errors than hourly collocation. A third protocol would be yearly collocation which is seldom used these days in model evaluation as it yields large representation errors (S16b). However, if remote sensing observations are used to construct a yearly climatology, effectively a yearly collocation protocol is used.

In data assimilation the representation error is often (but not always) thought to include effects from incorrectly modelled sub-grid processes. In this paper, the representation error is purely thought of as resulting from the different sampling by observations and models.

Section 2 describes the high-resolution simulation data and AERONET observations used in this study. The OSSE for estimating representation errors is briefly explained in Sect. 3 but more details can be found in S17. An evaluation of the high-resolution simulation with a particular focus on its use in an OSSE is given in Sect. 4. While the simulation shows deviations from AERONET observations, the agreement is deemed sufficient to study representation errors. Representation errors in AERONET AOT & AAOT are studied in Sect. 5. A ranking of AERONET sites in terms of their representativity is given in Sect. 6. As may be expected, the paper finishes with a summary of the conclusions (Sect. 7).

## 2   Data

### 2.1   GEOS-5 Nature Run

The GEOS-5 Nature Run (G5NR here-after) is a 2-year global, non-hydrostatic simulation from June 2005 to May 2007 at a $0.0625^o$ grid-resolution ($\sim 7$ km near the equator). Not just a simulation of standard meteorological parameters (wind, temperature, moisture, surface pressure), G5NR includes tracers for common aerosol species (dust, seasalt, sulfate, black and organic carbon) and several trace gases: $O_3$, $CO$ and $CO_2$. The simulation is driven by prescribed sea-surface temperature and sea-ice, daily volcanic and biomass burning emissions, as well as monthly high-resolution inventories of anthropogenic sources (Putman et al., 2014). As it is a nature run (i.e. no meteorological nudging), the meteorology in G5NR can deviate substantially from the actual weather in 2006.

Aerosol in GEOS-5 are calculated using the Goddard Chemistry, Aerosol, Radiation, and Transport (GOCART) module (Chin et al., 2002) that uses 15 tracers to describe externally mixed species of organic carbon, black carbon, sulphate, sea-salt and dust. Biomass burning emissions are obtained from QFED (Quick Fire Emissions Dataset) (Suarez et al., 2013) with a diurnal cycle imposed on-line. Anthropogenic emissions of organic and black carbon use EDGAR-HTAP (Emissions Database for Global Atmospheric Research-Hemispheric Transport of Air Pollution) emissions (Janssens-maenhout et al., 2012) which were rescaled to match AEROCOM Phase II emissions. Non-shipping anthropogenic $SO_2$ emissions come from EDGAR v4.1.

Evaluation of G5NR (Gelaro et al., 2015) against NASA/GMAO MERRA (Modern-Era Retrospective analysis for Research and Applications) Aerosol Reanalysis (da Silva et al., 2012) suggest that global organic carbon, black carbon and sulphate AOT are underestimated by $30-40\%$ while dust AOT is overestimated by $\sim 50\%$. Global sea-salt AOT is similar to MERRA within $10\%$. Hence, Castellanos et al. (2019) derived global rescaling factors for aerosol speciated AOT in G5NR but these are not used in the current

study (true scaling factors are unlikely to be global, it is unclear what to do about AAOT and the focus here is on relative errors anyway). Comparison with AEROCOM models shows that G5NR sulphate life-times are quite low (at 2.7 days) while the other species fairly agree with the AEROCOM multi-model mean. G5NR shows reasonable cloud fractions compared to CERES-SSF (Clouds and the Earth's Radiant Energy System-Single Scanner Footprints), although in the equatorial/sub-tropical region (30S-30N), G5NR has a deficit of partially cloudy scenes. In addition there are too few clouds off western continental coasts and the southern branch of the ITCZ is too strong. CALIOP (Cloud-Aerosol Lidar with Orthogonal Polarization) data suggests G5NR cloud fraction are too low, especially over equatorial/sub-tropical lands in the Northern Hemisphere, and too high in the northern polar region.

For this study, the following hourly G5NR data for 2006 were obtained: see Table 1.

**Table 1.** G5NR data used in this study

| short name | description |
|------------|-------------|
| totexttau | aerosol total column extinction at 550 nm |
| totscatau | aerosol total column scattering at 550 nm |
| swtdn | TOA* downward short-wave radiation |
| cldtot | total cloud area fraction |
| phis | surface geopotential height |
| bceman | monthly anthropogenic burning BC emissions |
| bcembb | monthly biomass burning BC emissions |

*: Top Of Atmosphere

## 2.2 AERONET observations & geolocations

AERONET (Holben et al., 1998) data were obtained from https://aeronet.gsfc.nasa.gov. For 2006, AOT from Direct Sun Version 3 L2.0 (Giles et al., 2019; Smirnov et al., 2000) and AOT & AAOT from Inversion Version 2 L1.5 and L2.0 (Holben et al., 2006) were logarithmically interpolated to values at 550 nm and averaged over an hour. For all years starting in 1992, geolocation data were obtained for all sites (1144 in total).

The DirectSun dataset contains only AOT (at multiple wavelengths). These observations are based on direct transmission measurements of solar light and have high accuracy of $\pm 0.01$ (Eck et al., 1999; Schmid et al., 1999). The Inversion dataset contains both AOT and AAOT (at multiple wavelengths) and these observations are based on measurements of scattered solar light from multiple directions. This inversion uses radiative transfer calculations (Dubovik and King, 2000) and yields larger errors than the DirectSun measurements. In particular, Dubovik et al. (2000) showed that Single Scattering Albedo (SSA) errors decrease with increasing

AOT and estimated SSA errors of $\pm 0.03$ for water-soluble aerosol at AOT at 440 nm $\geq 0.2$ and for dust or biomass burning aerosol at 440 nm $\geq 0.5$. Consequently, one important distinction between Inversion L1.5 and L2.0 data is a minimum threshold of AOT at 440 nm $\geq 0.4$ used in the latter (improved cloud screening is another distinction). Inversion L2.0 is a subset of the L1.5 dataset. For an intercomparison of AERONET SSA with flight campaign data, see Schafer et al. (2014).

In the current study, only AOT at 550 nm is used and the Inversion L2.0 AOT at 440 nm criterion is adapted to AOT at 550 nm $\geq 0.25$. This is the minimum value of AOT at 550 nm present in actual Inversion L2.0 data, but also corresponds to AOT at 440 nm = 0.4 for small particles (Ångström exponent = 2.1). As a result, the OSSE in this paper is rather lenient when it comes to selecting valid observations similar to Inversion L2.0.

## 2.3 GAW geolocations

GAW geolocation data were obtained from NILU (Norwegian Institute for Air Research). Two networks were used: the GAW-AOT network which comprises 29 sun-tracking sun photometers that measure AOT; and the GAW-ABS network which comprises 81 filter instruments that measure surface properties. The real GAW-ABS network is not capable of measuring a columnar (A)AOT but here we will assume it does, similar to AERONET, and consider its representation errors.

## 3 Method: analysis of representation errors

The representation error is defined as the difference between a perfect observation (i.e. no observational error) and a truth value (area average), see also S16a and S17. Here, a self-consistent high resolution simulation will be used to generate both observation and truth in a so-called Observing Systems Simulation Experiment. The representation error may refer to instantaneous values or time averages. This work concerns itself mostly with yearly averages (and some monthly averages). For instantaneous and daily error values, see S16a and S17. The mapping from G5NR data to the data used in this study is given in Table 2.

Perfect observations are generated from the high-resolution simulation by choosing the data at the location of an AERONET or GAW site and sub-sampling those data in time according to certain conditions for solar zenith angles (SZA), cloud-fraction and AOT. Table 3 lists the threshold conditions for which observations will be possible. Values for SZA and AOT are inferred from real AERONET data files. The maximum cloud-fraction was tuned to obtain similar temporal coverage of observations as real AERONET data (see Sect. 4 and Fig. 3 but the impact of tuning is small).

**Table 2.** Mapping from G5NR data to data used in this study

| G5NR | this study | units |
|---|---|---|
| totexttau | AOT | |
| totextau-totscatau | AAOT | |
| $\frac{180}{\pi}\arccos(\text{swtdn}/1367)$ | SZA | degrees |
| cldtot | cloud fraction | |
| phis/9.81 | geopotential altitude | m |
| bceman+bcembb | BC emissions | kg/m$^2$ s |

**Table 3.** Conditions for valid AERONET observations as simulated in this study

| source | maximum SZA | maximum cloud-fraction | minimum AOT |
|---|---|---|---|
| DirectSun L2.0 | $80^o$ | 0.01 | 0.0 |
| Inversion L1.5 | $80^o$ | 0.01 | 0.03 |
| Inversion L2.0 | $80^o$ | 0.01 | 0.25 |

The truth is generated from the high-resolution simulation by averaging AOT and AAOT over a large area ($0.5^o$ to $4^o$ grid-boxes) and further averaging in time. Here we should distinguish three different protocols depending on how one [5] intends to use the observations, see Table 4. In the case of a gridded climatology derived from observations, the truth should be an average over a continuous long-term time range (say a year). In the case of model evaluation, it is possible to resample model data to the times of the observations. E.g. [10] within the AEROCOM community, a daily collocation protocol is often used, where daily model data is used for days with observations only (irrespective of the temporal sampling of those observations throughout the day). To assess representation errors in this case, the truth needs to be sampled [15] accordingly to days with observations before yearly averages are determined. The same protocols were also explored in S17.

**Table 4.** Collocation protocols

| collocation protocol | purpose |
|---|---|
| yearly | gridded climatology |
| daily | model evaluation (AEROCOM) |
| hourly | model evaluation |

The current methodology differs slightly from S17 in that:

1. a different model is used to construct the OSSE,

2. previously, SZA was assumed to be sufficiently high for [20] a fixed fraction of the day (10 hours). In the current work, SZA is calculated from downward-welling TOA SW radiation and will vary with geo-location and time-of-day,

3. previously, the truth was generated for grid-boxes [25] centered on the observations. In the current work, those grid-boxes are assumed regularly spaced from $0^o$ to $360^o$ longitude and $-90^o$ to $90^o$ latitude. The AERONET and GAW sites can be located anywhere within those grid-boxes (at their real geo-location), [30]

4. previously, the high-resolution simulation had a constant grid-size of (about) 10 km. In the current work, the grid-size varies but has a constant angular size of $0.0625^o$ ($\sim 7$ km at the equator).

The last point implies that the simulation grid-box used for [35] the observation decreases towards zero as we approach the poles. Since this is clearly undesirable (field-of-view will remain on the order of several kilometers), we will limit our analysis of *representation errors* to latitudes below $60^o$. The exception is Fig. 5. [40]

Our methodology allows separation of the factors that determine the representation error: spatial extent of the grid-box, and observational intermittency due to low SZA, high cloud-fraction or low AOT. We will not present such *causal analysis* in this paper (see S17 instead) but will refer to it to [45] explain results.

To show the distributions of representation errors, box-whisker plots using the 2, 9, 25, 75, 91 and 98% quantiles will be used in this paper (in addition, the median is shown as a bar and the mean as a circle). For a normal dis- [50] tribution, these quantiles will be equally spaced. Any skewness or extended wings in a distribution will be readily visible. In addition to quantiles, the values of mean error and the mean sign-less error will be provided. The mean sign-less error is deemed more relevant than the standard devi- [55] ation as 1) it includes biases; 2) the errors are seldom normally distributed, and a standard deviation is very sensitive to larger errors ("out-liers"). For a normal distribution with a mean of zero and a standard deviation of one, the mean sign-less error is $\sim 0.8$. The correlation used in this paper is [60] the Pearson correlation coefficient that assesses linear relationships. Regression slopes were calculated with a robust Ordinary Least Squares regressor (OLS bisector from the IDL `sixlin` function, Isobe et al. (1990)). This regressor is recommended when there is no proper understanding of [65] the errors in the independent variable, see also Pitkänen et al. (2016).

## 4 Evaluation of G5NR and OSSE

In this section, G5NR is evaluated with real AERONET observations of AOT and AAOT, with special focus on its use- [70]

fulness in an OSSE. As G5NR generates its own meteorology that deviates from 2006, one might expect differences between simulation and observations. Simulated data were nevertheless collocated to the time of the observations (within the hour) to ensure the same temporal sampling throughout the days, the months and the year.

The mean and standard deviation of AOT and AAOT per site are shown in Fig. 1, top row. In general, simulated site-mean AOT shows good agreement with the observations with correlations around 0.75 and slopes around 0.84. Simulated site-mean AAOT does not agree as nicely with the observations but there is still correlation (0.48) (the evaluation of AAOT will be affected by large measurement errors). The agreement in standard deviation suggests that simulated and observed AOT (and AAOT) show similar temporal variation. But the global agreement also suggests that the simulation captures spatial variation rather well. This is also true on shorter length scales, as an analysis by region shows in Table 5. Europe appears to be the exception but this is mostly due to a few southern sites. As the table shows: without those sites, correlation increases significantly. This may be related to the overestimation of dust and underestimation of carbonaceous & sulphate aerosol in G5NR (Gelaro et al., 2015), which will affect north-south gradients in AOT in Europe. DRAGON (Distributed Regional Aerosol Gridded Observation Networks, see Holben et al. (2018)) campaigns might allow evaluation of the spatial distribution of simulated AOT at even smaller length-scales (10's of kilometers) but are not available for 2006.

**Table 5.** Correlation in modelled and observed yearly site-mean AOT

| region | nr | correlation |
|--------|-----|-------------|
| World | 216 | 0.75 |
| Europe | 55 | 0.26 |
| Europe* | 26 | 0.68 |
| Africa | 32 | 0.86 |
| Asia | 34 | 0.82 |
| N. America | 49 | 0.81 |
| S. America | 13 | 0.91 |

*: southern AERONET sites removed from analysis

The top row of Fig. 1 was created using only sites that provide a minimum of 100 real observation throughout 2006. The lower row shows how this criterion affects results. As the minimum number of observations per site increases, so do the correlations, probably due to a reduction in statistical noise (partly due to different simulated and actual meteorologies). But the overall bias also increases. This criterion selects for sites with lower cloudiness (higher number of observations) until predominantly northern African and Saudi Arabian sites are left for a minimum of 500 observations per site. The increase in bias is thus likely due to the overestimation of dust AOT that was mentioned earlier.

Note that AAOT is here evaluated with L1.5 data. The L2.0 data have a minimum AOT threshold which results in fewer observations and fewer available sites overall. Although L1.5 is considered a less reliable product, the evaluation with L2.0 (which now uses a minimum of 30 observations per site) yields a similar but slightly poorer result for G5NR, see Fig. S1, and over a shorter range of values.

Figure 2 shows mean values per site for the daily difference in maximum and minimum AOT. Again, good agreement for simulated AOT is seen but AAOT compares rather poorly. However, its correlation is still above 0.6 and it is clear that the simulation *underestimates* daily AAOT variation. The impact of AAOT measurement error on daily variation is likely reduced as the variation is a difference between two measurements (pers. comm. with T. Eck and O. Dubovik).

Figure 3 shows the temporal coverage (or frequency of observation) per site as a function of latitude. G5NR's simulated coverage is calculated using the conditions described in Table 3 (and explained in Sec. 3). This coverage would be 100% if observations are available 24 hours a day, 365 days a year. In practice it cannot be higher than 50% due to the day-night cycle, and will be less due to cloudiness or low AOT.

The bimodal structure that is visible in both the simulation and observations is due to SZA variation (which reduces coverage towards the poles) and cloudiness (which reduces coverage near the equator). Simulated and real coverage per site are not expected to agree well due to meteorological differences and down times from site maintenance. Still, the results suggests that the OSSE predicts similar frequency of Direct Sun observations as actually observed.

However, the OSSE also simulates more Inversion observations in the Northern hemisphere than actually occur. This suggests there are additional factors in observational coverage that are not accounted for in Table 3. One factor is that real Inversion measurements are attempted less frequently (several times per day) than Direct Sun measurements (several times per hour). Other factors may include inversion failure at low SZA (real observations show that Inversion data generally have larger SZA than DirectSun data even though Inversion data is generally closer to the equator) and overestimation of dust AOT in G5NR (largest over-estimates of coverage occur for Sahara and Saudi Arabia sites). Yet another issue is that successful inversion requires a high degree of azimuthal symmetry in the measurements. In essence, this is a built-in check on the magnitude of *spatial* representation errors which will lower temporal coverage of the observations but is not considered in the OSSE due to lack of information. Finally, instrument malfunction & maintenance are not taken in to account, although that would affect AOT coverage as well.

In all, it seems that G5NR can realistically simulate spatial and temporal variation in AOT and AAOT, at least on the scales accessible by the available observations. There is some underestimation of daily AOT variation and significant underestimation of daily AAOT variation. G5NR can also be used to fairly realistically simulate frequency of observation (temporal coverage), although it will over-estimate this for the Inversion products in the Northern Hemisphere.

## 5  Results

### 5.1  Representation errors in yearly AOT

Figure 4 shows yearly representation errors for AERONET DirectSun L2.0 AOT observations as a function of model grid-box size, for the three collocation protocols (see Table 4). As grid-box size changes from $4^o$ to $0.5^o$, errors for hourly collocation are more than halved from 13% to 5% while those for daily collocation change only from 17% to 12%. In contrast, errors for yearly collocation ($\sim 22\%$) are dominated by temporal sampling and do not depend much on grid-box size. Smaller representation errors for hourly collocation can also be seen in a regional analysis, see Fig. S2. The hourly collocation is especially beneficial when using the Inversion L2.0 AOT product, which allows large representation errors due to the condition of a minimum AOT at 440 nm ($>= 0.4$) for valid observations, see Fig. S3, even though it results in a global 9% bias.

The impact of collocation protocol can also be shown through the total number of sites that yield errors larger than, say, 10%: 821 (yearly), 653 (daily), 235 (hourly) out of 1108 AERONET stations in total, for a grid-box of $1^o \times 1^o$.

In addition to larger representation errors in general, the yearly and daily collocation protocols also allow significant biases across the AERONET network. Regionally, spatial patterns with east-west or north-south gradients in the representation errors exist, see Fig. 5 and Fig. S4. Such patterns are absent or at least much reduced for hourly collocation.

The biases in regional and global distributions of representation errors for yearly and daily collocations are strongly affected by cloudiness. Higher humidity in the cloudy part of a grid-box increases AOT through hygroscopic growth The area averages used to calculate representation errors have been derived for the *entire* grid-box (all-sky), both clear and cloudy parts. Representation errors for clear-sky parts of grid-boxes are lower for the yearly and daily collocation protocols, see Fig. 6. In certain situations, it seems more realistic to use only the clear part of the grid-box in calculating representation errors: e.g. when the grid-box average stands in for an aggregated satellite product. In this paper, focus will be on the all-sky representation error.

Table 6 shows absolute values of the yearly representation errors for different collocation protocols (yearly and hourly) and grid-box sizes. The statistical metrics provided

are the mean of the sign-less representation error over all AERONET sites, and the 90% quantile of the sign-less representation error (an indication of the large representation errors possible for some sites). Using absolute values allows a comparison with the AERONET AOT measurement error of 0.01 (Eck et al., 1999; Schmid et al., 1999). This is the error for individual measurements, and not that of a yearly average which is likely to be much smaller. Clearly, representation errors are larger than measurement errors.

Results so far suggest that the daily collocation is a significant improvement over the yearly collocation. This is in contrast to S17 (Fig. 7) where the representation errors for daily and *monthly* collocation were found to be similar. The absence of diurnal (anthropogenic) emission profiles in G5NR may cause underestimation of representation errors for the daily collocation in the current study.

It is interesting to compare the representation errors of two different networks, AERONET and GAW. AERONET was not designed with representativity in mind but the GAW network was. Nevertheless, Fig. 7 suggests that GAW sites exhibit slightly larger representation errors than AERONET. In particular, GAW error statistics are strongly skewed to negative values. In the G5NR OSSE, GAW sites are located at higher altitudes and more often on isolated mountains than AERONET sites (G5NR site altitudes correlate very well with real altitudes, $R = 0.98$, but tend to underestimate by 28 m on average, with a standard deviation of 171 m). A look at yearly representation errors for the hourly collocation reveals a systematic altitude dependence, see Fig. 8. A high altitude site on an isolated mountain will observe a shorter atmospheric column than the surrounding grid-box (most of which is at lower altitudes) which will cause a negative representation error. Note that AERONET sites do not show this dependence on altitude for $1^o$ grid-boxes, probably because they are located more often on mountains surrounded by similar mountaineous terrain.

Finally, a comparison is made with a previous study into AERONET representation errors (Kinne et al., 2013). Using a range score $r$, see Table 7, they ranked sites according to their representativity for larger domains. This ranking is subjective in that it is non-quantitative, based on personal knowledge of the sites and only defines representativity in broad terms. The range scores are only available for sites that had at least 5 months of data before 2008. Using the methodology of this paper, representation errors were calculated for all sites of a certain range score, see Fig. 9. For large grid boxes of $4^o$ ($\sim 450$ km near equator), the impact of the range score on representation error is quite small. While there is a visually arresting change in the error distribution for $r > 1$ (wide flanks are changed into a broader center), the mean sign-less error barely changes. This rather weak dependence on range score suggests that Kinne et al. (2013) overestimated the size of the domains ($>= 500$ km for $r > 1$) for which their sites were representative. On the other hand, for a grid-box of $1^o$ a substantial reduction in representation error can be seen

**Table 6.** Absolute representation errors for AERONET sites

| metric | protocol | $4^o$ | $2^o$ | $1^o$ | $0.5^o$ | $0.0625^o$ |
|--------|----------|-------|-------|-------|---------|------------|
| mean | yearly | 0.043 | 0.042 | 0.042 | 0.042 | 0.044 |
|  | hourly | 0.021 | 0.015 | 0.011 | 0.008 | 0.000 |
| 90 % quantile | yearly | 0.086 | 0.079 | 0.082 | 0.083 | 0.086 |
|  | hourly | 0.052 | 0.033 | 0.029 | 0.017 | 0.000 |

for $r \geq 1$ sites. However, this only occurs for the hourly collocation: Kinne et al. (2013) did not consider the temporal sampling of the observations which causes large representation errors. An alternative ranking of representativity will be introduced in Sect. 6.

## 5.2 Representation errors in monthly AOT

Surprisingly, monthly representation errors are not that much larger than yearly errors, see Fig.10. If monthly errors for the same site were independent and random, one would expect them to be $\sim \sqrt{12} \approx 3.5$ larger than yearly errors but that is not the case. As a matter of fact, monthly errors are strongly correlated from month to month, throughout the year, see Fig. 11. The increase in correlation with January after September, is probably due to yearly cycles in meteorology and emissions and very likely to be a realistic aspect of representation errors. The implication of this is that multi-year averages may not reduce representation errors as strongly as one would hope.

This analysis also provokes the question whether representation errors (per site) should be seen as mostly biases or random errors (see also Schwarz et al. (2018)). Preliminary analysis suggests that both cases can occur. I.e. some sites show large variations (including sign changes) in representation error from month to month, and as a consequence a strongly reduced yearly representation error. Here monthly representation errors may be interpreted as mostly random. Other sites show monthly representation errors with not much variation and as a consequence yearly representation errors are similar to the monthly errors. There the representation error is better characterised as a bias. A proper analysis of this would require significantly longer time-series of data than are currently available. Further discusion of this can be found in Sect. 6.

## 5.3 Representation errors in AAOT

The discussion of representation errors for Inversion L1.5 AAOT will be shorter than that for DirectSun L2.0 AOT, as the main conclusion is identical: the hourly collocation yields smaller representation errors than the other protocols, see Fig. 12. Note also that representation errors in AAOT are of a similar magnitude as for AOT. One obvious difference is that AAOT representation errors tend to be positively biased while the AOT errors were negatively biased. While the latter was due to cloudiness as discussed before, the positive bias for AAOT is more difficult to explain. It appears that a combination of conditions (location of the sites, necessity of day-light, clear skies and a minimum AOT of 0.03) together conspire to create these positive biases. Only over the Amazon can a simple explanation be found: the clear sky condition prevents many observations outside the biomass burning season, explaining large positive biases for yearly collocation (see also Fig. S5, discussed later).

Even more than for AOT, representation errors for AAOT are very similar for the daily and hourly collocations. As discussed before, this is likely due to the absence of diurnal (anthropogenic) emissions profiles. The daily variation of AAOT is strongly underestimated by G5NR (see Sect 4 and Fig. 2).

For completeness' sake, an analysis of AAOT representation errors for different regions (Fig. S5), different products (Fig. S6), different networks (Fig. S7) and different range scores by Kinne et al. (Fig. S8) are given in the supplement. Overall the conclusions are very similar to those for AOT.

The similarity in general behaviour of representation errors for AOT and AAOT should not be taken to mean that these errors are identical per site. As discussed in Sect. 6, representation errors for AOT and AAOT at individual sites can be very different. Ultimately this is due to the different sources of AOT and AAOT which leads to different spatio-temporal distributions in the atmosphere.

## 5.4 Comparison to recent results from Wang et al. '18

Recently Wang et al. (2018) suggested that the observed underestimation of AAOT by AEROCOM models (Bond et al., 2013) may be due to *spatial* representation errors. Spatial representation errors are entirely due to the narrow field-of-view of AERONET observations (i.e. the intermittent temporal sampling of these observations is ignored). Their analysis found that AERONET Inversion L1.5 AAOT representation errors exhibit a global bias of $30\%$ for $2^o \times 2^o$ model grid-boxes, which would help explain the aforementioned underestimation by the global models. As AERONET sites need to be serviceable, they are often found near roads and ur-

**Table 7.** Range scores for AERONET sites in (Kinne et al., 2013)

| range score | spatial domain | number of sites | comments |
|---|---|---|---|
| 0 | 100 km | 120 | includes mountainous sites |
| 1 | 300 km | 106 | |
| 2 | 500 km | 28 | |
| 3 | 900 km | 6 | |

ban build-up, i.e. near sources of absorbing aerosol. Compared to the larger area of global model grid-boxes, these sites would quite naturally observe larger AAOT. Thus, Wang et al. (2018) concluded that at least part of the underestimation of modelled AAOT is an artefact, created by the location of the AERONET sites.

Wang et al.'s idea is quite persuasive and indeed one can see evidence of such positive representation errors in Fig. 13 where sites in major cities like London, Paris, Madrid and Barcelona clearly exhibit positive representation errors. (For another example, see Fig. 3b in S17 concerning surface black carbon concentrations). But Wang's study found such biases for the majority of AERONET sites, not just a few located in big cities. As a matter of fact, the current study shows no evidence of this global bias of 30%. Instead it finds a global bias of only 9%, dominated by a few sites with large positive representation errors (median bias over all sites: 4%).

Wang et al. (2018) performed an analysis very much like the one in this study with one crucial difference. As they did not have a global simulation at high resolution like G5NR, they downscaled results from a standard global simulation at $2.5^o \times 1.27^o$ resolution. The downscaling was accomplished with the help of a high-resolution ($0.1^o \times 0.1^o$) black carbon emission map (Wang et al., 2016). It is possible to simulate this procedure using the high-resolution G5NR black carbon emission maps and AAOT simulations (the AAOT simulation was first coarsened over $2^o \times 2^o$) and explain the different results in Wang et al. (2018) and the current study.

Figure 14 shows AAOT *spatial* representation errors as estimated by the current study and by Wang's methodology as simulated with G5NR data. A global bias of 25%, not very different from the original 30% mentioned in Wang et al. (2018), is found for the Wang analysis whose representation errors yield a strongly skewed distribution over all sites. In contrast, the present study yields a more symmetric distribution with a much smaller bias. Unlike in the Wang analysis this bias is dominated by just a few sites with large positive representation errors.

The analysis above is a self-consistent evaluation of Wang's methodology. Using high-resolution black carbon emission data to downscale coarse model AAOT fields ignores redistribution of absorbing aerosol due to small scale (at and below the coarse model's grid-box) advective and turbulent transport as well as removal by local precipitation (Wang et al. were aware of this limitation but could not assess its impact). It also ignores local orography and the contribution of absorbing dust to AAOT. The result is that there is very little correlation between representation errors as estimated by the two methods, see Fig. 15. As a matter of fact, representation errors from the current study do not show a systematic dependence on emission distributions, unlike the representation errors from Wang's methodology.

## 6   A ranking of representativity for the AERONET sites

A ranking of AERONET and GAW sites in terms of their *spatial* representativity for AOT and AAOT can be found at Schutgens (2019). Only sites below $60^o$ latitude are considered, and temporal sampling of observations is ignored. The latter was done for two reasons: 1) as discussed in Sect 2 and 4, temporal sampling of observations is considered less accurately modelled by the OSSE than spatial variability; 2) both S17 and the current study show that once hourly collocation is used, the remaining representation error is similar although slightly larger than the spatial representation error.

Relative representation errors are classed according to bins: 0-5% (rank 1), 5-10% (rank 2), 10-20% (rank 3), 20-40% (rank 4), 40% and higher (rank 5). The accuracy of this ranking depends of course on the skill of G5NR and the OSSE, but also on statistical noise due to the use of a single year of data. The latter source of uncertainty was assessed using a block bootstrap method (Efron, 1979) on the time-series per site. Typically more than 85% of all resampled time-series yield a representation error in the same class as the original time-series. For large grid-boxes ($4^o$) and small errors ($< 10\%$), this may drop down to 66% of the resampled time-series. For those resampled time-series that yielded a different ranking, this ranking was only off by 1. It then seems that statistical noise does not prevent a robust classification of yearly relative spatial representation errors. The impact of G5NR and OSSE skill on the classification can currently not be assessed.

Compared to the subjective ranking by Kinne et al. (2013), the new ranking is objective because the rank is related to a well-defined representation error that is quantified bottom-up

from known emission sources and calculated meteorology. That in itself is of course no guarantee for accuracy.

Inspection of the rankings turns up several interesting points. Analysis in the previous sections determined a few "rules" for the behaviour of representation errors (e.g. errors decrease when the grid-box size decreases) but these can easily be "broken" for specific sites: a smaller grid-box may actually lead to larger representation errors (e.g. AOE_Baotou, Ascension_Island, Aras_de_los_Olmos), monthly errors may be substantially larger than yearly errors (e.g. ARM-Darwin, BORDEAUX). Also, representation errors for AOT and AAOT may be very different: Bayfordbury shows small yearly representation errors for AOT but large errors for AAOT, while Mace_Head shows the opposite.

## 7  Conclusions

Remote sensing observations from the AERONET and GAW networks are intermittent in time and have a limited field-of-view. Consequently such observations have limited ability to represent (Absorbing) Aerosol Optical Thickness, or (A)AOT, over larger areas. The resulting spatio-temporal *representation* error is here analysed using a high-resolution simulation of global aerosol (GEOS5 Nature Run, $\sim 7$ km resolution near equator). Using G5NR, an Observing System Simulation Experiment (OSSE) was constructed that simulates the frequency of AERONET observations taking Solar Zenith Angle, cloud fraction and AOT values into account.

This work extends previous work on temporal representation with global low-resolution models (Schutgens et al., 2016b) to spatio-temporal representation. It also extends previous work on spatio-temporal representation with regional high-resolution simulations (Schutgens et al., 2016a, 2017) to the global domain. The current work is more limited in scope than the previous studies and only considers ground-based remote sensing observations. For satellite remote sensing, see Schutgens et al. (2016b) and Schutgens et al. (2017). For in-situ measurements, see Schutgens et al. (2016a) and Schutgens et al. (2017).

G5NR and the OSSE are evaluated and found to show significant skill in AOT and reasonable skill in AAOT. AERONET mean AOT per site, as well as yearly and daily variability were estimated quite correctly, usually within a factor less than $2\times$. Considering that G5NR generates its own meteorology, G5NR AOT correlated very well ($R \approx 0.75$) with the observations. Similarly, the OSSE was surprisingly good at simulating the overall pattern of observational coverage (frequency of AOT observation). Results were not as good for AAOT but still acceptable. Yearly AAOT variability was slightly underestimated while daily AAOT variability was severely underestimated. The latter is possibly related to the absence of diurnal anthropogenic emission profiles in G5NR. For representativity studies that take diurnal variations into account, see Schutgens et al. (2016a, 2017).

In addition, the OSSE tended to overestimate the frequency of AAOT observations per site (although this was shown to have no impact on representation errors).

Both yearly and monthly representation errors are provided for observations from ground sites that attempt to represent larger areas (from $0.5^o$ to $4^o$ in size). The monthly representation errors are shown to be strongly correlated throughout the year. For some sites this is an expression of a bias but that is not universally the case. In any case, monthly representation errors can not be treated as independent and this has (negative) consequences for the reduction of representation errors in multi-year averages. Other conclusions are: 1) AERONET derived climatologies allow for substantial representation errors (yearly collocation allows errors of typically 20%, see Fig. 4); 2) AEROCOM evaluation protocol is sub-optimal (daily collocation can show errors of 25% in coherent regional patterns). Instead hourly collocation is advocated, 3) the representativity of AERONET and GAW sites was shown to be not very different, although AERONET sites seem to be more affected by nearby sources while GAW sites seem more affected by their altitude. Finally, a subjective ranking (Kinne et al., 2013) of the *spatial* representativity of sites was analysed and shown to broadly agree with the current study, although it appears to overestimate represented spatial domain sizes and judges several sites as less representative than the current analysis. A new objective ranking is also presented.

While the current study's focus is on strategies for model evaluation with original ('All Points') AERONET data, it does allow recommendations to be made for the optimal aggregation level of observational data. Hourly products are preferred to daily or monthly products as they allow users to perform hourly collocation which in turn yields significantly smaller representation errors. This should hold for both satellite and AERONET data.

*Spatial* representation errors have been used to reconcile observations and global simulations of AAOT. Bond et al. (2013) showed that global models tend to significantly underestimate AAOT but Wang et al. (2018) suggested that AERONET AAOT observations may suffer from a global 30% representation bias. In contrast, the current analysis finds a much smaller bias of $9\%$ which is more-over strongly influenced by a few sites with large positive representation errors due to their proximity to black carbon sources. Judiciously excluding those sites significantly reduces the bias even further ($4\%$). The large positive representation errors found by Wang et al. are shown to be due to methodological choices that limit the realism of their OSSE.

Several questions remain and seem interesting for follow-up studies: 1) how can we evaluate the representativity rankings; 2) how do OSSE errors affect estimated representation errors; 3) how will diurnal emission profiles impact results; 4) can representation errors at any site be decomposed in a bias and random error (possibly with temporal correlations

over several months); 5) what are representation errors like in multi-year averages?

*Code and data availability.* G5NR data can be obtained from `https://gmao.gsfc.nasa.gov/global_mesoscale/` `7km-G5NR/data_access`, AERONET data can be obtained from `https://aeronet.gsfc.nasa.gov`. Analysis code was written in IDL and is available from the author upon request.

*Author contributions.* NS designed the experiments, carried them out and prepared the manuscript.

*Competing interests.* No competing interests are present

*Acknowledgements.* NS thanks the NASA Global Modelling and Assimilation Office team that conducted the GEOS-5 Nature Run simulation, in particular Arlindo da Silva and Ravi Govindaraju for help in obtaining the G5NR data. NS thanks the PI(s) and Co-I(s) and their staff for establishing and maintaining the many AERONET sites used in this investigation. NS thanks Ann Mari Fjaeraa (NILU) for providing GAW-AOT and GAW-ABS geolocation data. NS is also grateful to Rong Wang, Björn H. Samset and Gunnar Myhre for valuable discussions. Three anonymous reviewers, Andrew Sayer and Tero Mielonen provided very useful commentary and NS gratefully acknowledges their contributions. The figures in this paper were prepared using David W. Fanning's Coyote Library for IDL. This work is part of the Vici research programme with project number 016.160.324, which is (partly) financed by the Dutch Research Council (NWO).

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

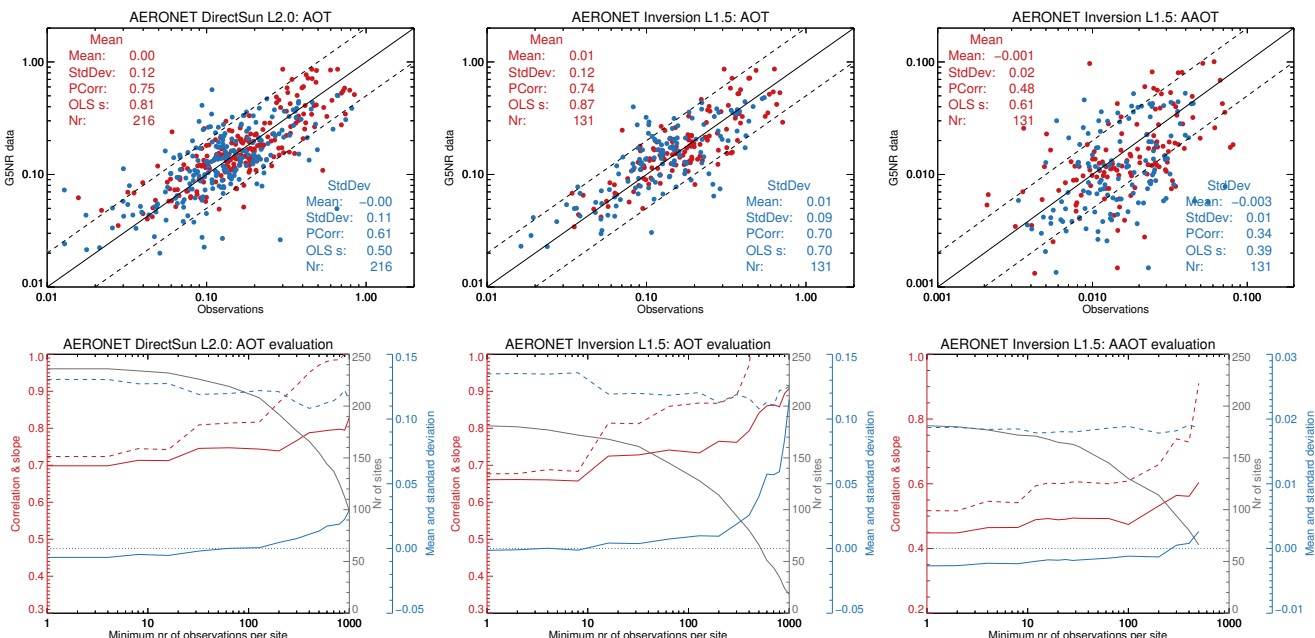

**Figure 1.** Evaluation of the G5NR simulation of AOT and AAOT with AERONET data. The top row shows evaluation against three different datasets. Each dot represents the yearly mean or standard deviation for a single AERONET site (with at least 100 observations in 2006); the mean value is shown in red and the standard deviation in blue. The coloured text summarizes the statistics over all data points in the figure. In the bottom row, the impact of the minimum required number of observations per site on those summary statistics (for means) is shown. Colours relate lines to axes and have different meaning than in the top row. Red solid is correlation, red dashed is slope, blue solid is mean, and blue dashed is standard deviation. In all figures, hourly G5NR model data was collocated in time & space with AERONET observations before calculating site statistics.

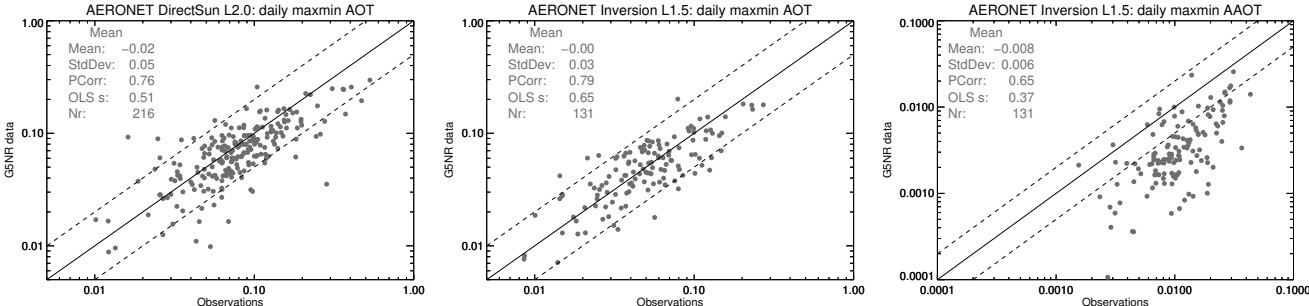

**Figure 2.** Evaluation of the G5NR simulation of AOT and AAOT with AERONET data. Each dot represents the yearly average of daily variation (maximum minus minimum value) for a single AERONET site (with at least 100 observations in 2006). The grey text summarizes the statistics over all data points in the figure. In all figures, hourly G5NR model data was collocated in time & space with AERONET observations before calculating site statistics.

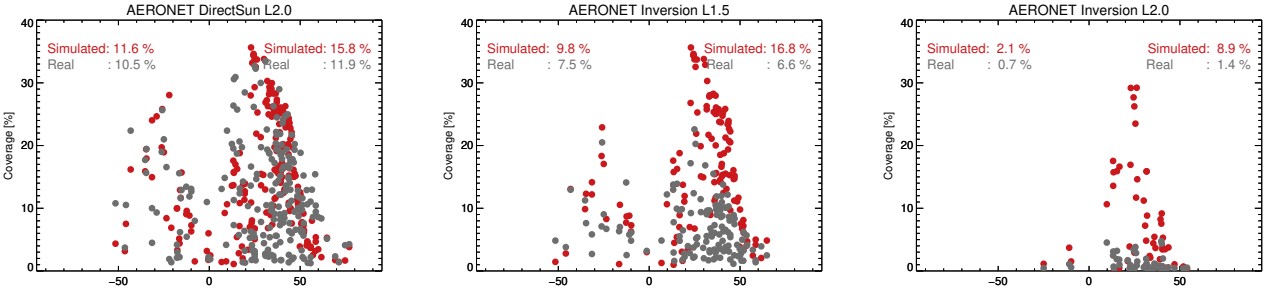

**Figure 3.** Evaluation of the temporal coverage predicted by the OSSE with AERONET observations. Each dot represents temporal coverage (or frequency of observation) for a single AERONET site (with at least 100 observations in 2006, at least 30 observations for Inversion L2.0). The grey dots are real AERONET data, the red dots are simulated by the methodology described in Sec. 3. The numbers in the graph are temporal coverages estimated by hemisphere.

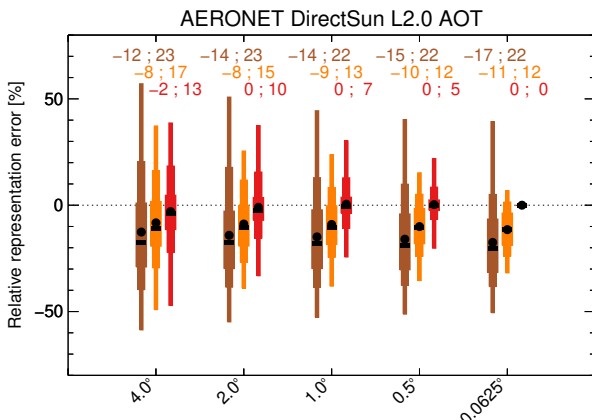

**Figure 4.** Yearly representation errors for AOT from DirectSun L2.0 AERONET for different model grid-box sizes. The colours indicate different collocation protocols: yearly (brown), daily (orange) and hourly (red). Numbers on top are mean of the errors and mean of the sign-less errors.

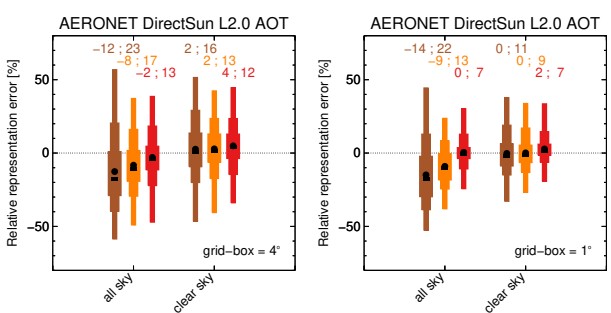

**Figure 6.** Yearly representation errors for AOT from from Direct-Sun L2.0 AERONET using all-sky or clear sky conditions and model grid-box size of $4^o$ (left) or $1^o$ (right). The colours indicate different collocation protocols: yearly (brown), daily (orange) and hourly (red). Numbers on top are mean of the errors and mean of the sign-less errors.

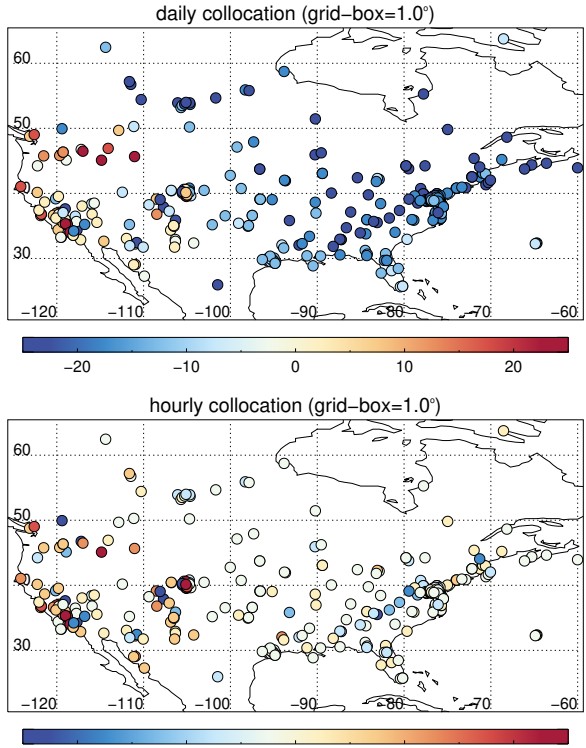

**Figure 5.** Yearly representation errors [%] for AOT from DirectSun L2.0 AERONET in Northern America, for two different collocation protocols (top: daily; bottom: hourly) and a model grid-box size of $1^o$.

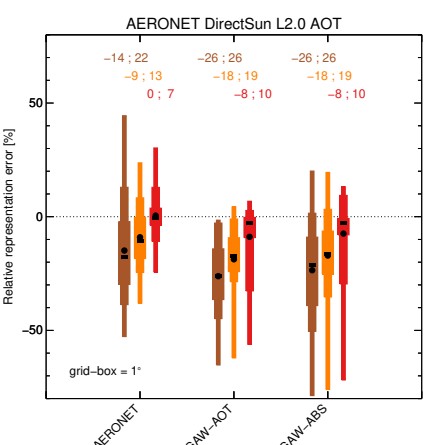

**Figure 7.** Yearly representation errors for AOT from Direct Sun L2.0 AERONET and GAW and a model grid-box size of $1^o$. The colours indicate different collocation protocols: yearly (brown), daily (orange) and hourly (red). Numbers on top are mean of the errors and mean of the sign-less errors.

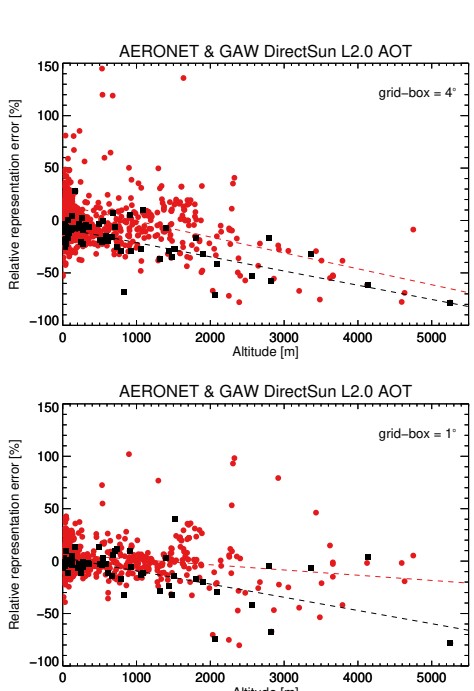

**Figure 8.** Yearly representation errors for AOT from Direct Sun L2.0 AERONET (red circles) and GAW (black squares) as a function of site altitude, for a model grid-box size of either $4^o$ or $1^o$; using hourly collocation.

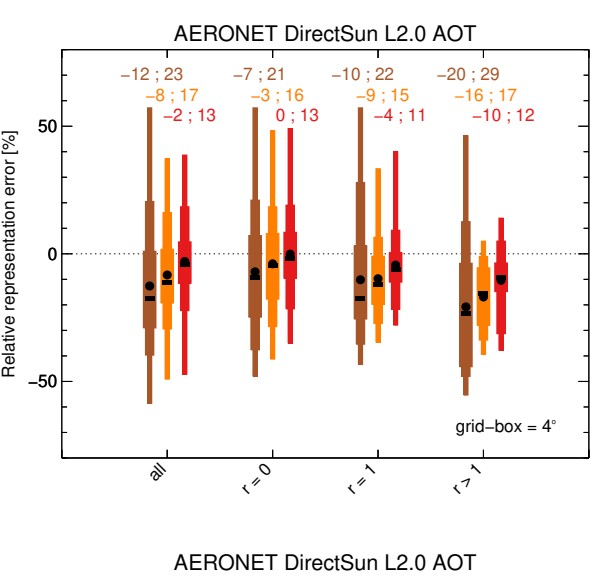

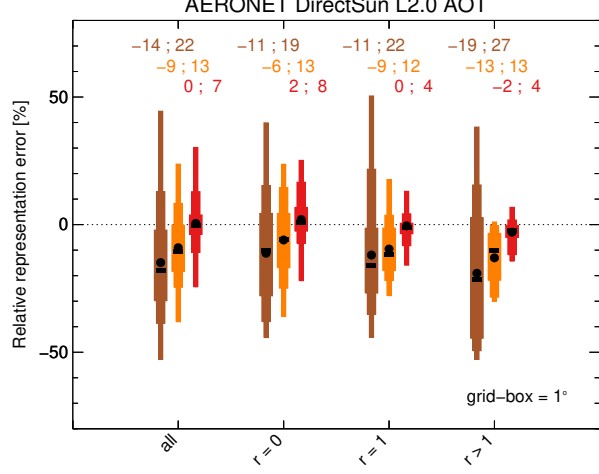

**Figure 9.** Yearly representation errors for AOT from DirectSun L2.0 AERONET for different range scores $r$ by Kinne et al. (2013), for a model grid-box size of either $4^o$ or $1^o$. The colours indicate different collocation protocols: yearly (brown), daily (orange) and hourly (red). Numbers on top are mean of the errors and mean of the sign-less errors.

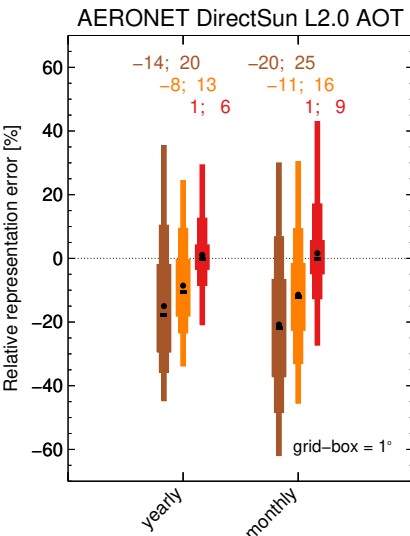

**Figure 10.** Yearly and monthly representation errors for AOT DirectSun L2.0 AERONET, for a model grid-box size of $1^o$. In contrast to Fig. 4, only sites that provide observations 12 months out of the year are used in this analysis. The colours indicate different collocation protocols: yearly (brown), daily (orange) and hourly (red). Obviously, for the brown bar on the right a monthly protocol was used. Numbers on top are mean of the errors and mean of the sign-less errors.

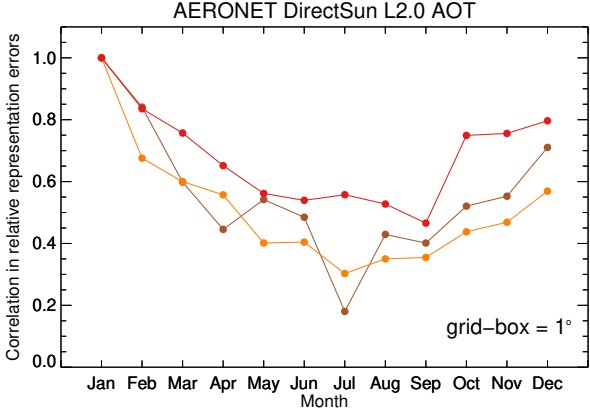

**Figure 11.** Correlation in monthly representation errors with errors for January, for AOT DirectSun L2.0 AERONET, for a model grid-box size of $1^o$. Only sites that provide observations 12 months out of the year are used in this analysis. The colours indicate different collocation protocols: monthly (brown), daily (orange) and hourly (red).

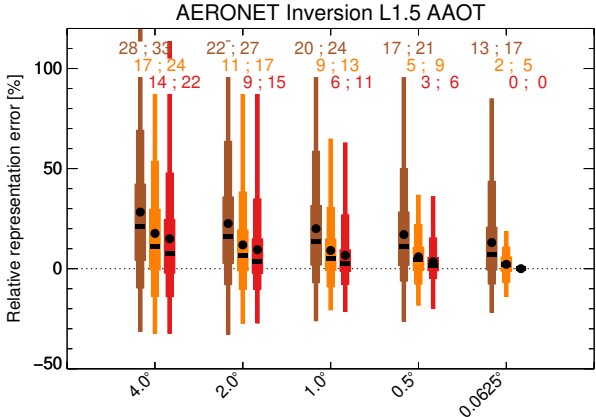

**Figure 12.** Yearly representation errors for AAOT from Inversion L1.5 AERONET for different model grid-box sizes. The colours indicate different collocation protocols: yearly (brown), daily (orange) and hourly (red). Numbers on top are mean of the errors and mean of the sign-less errors.

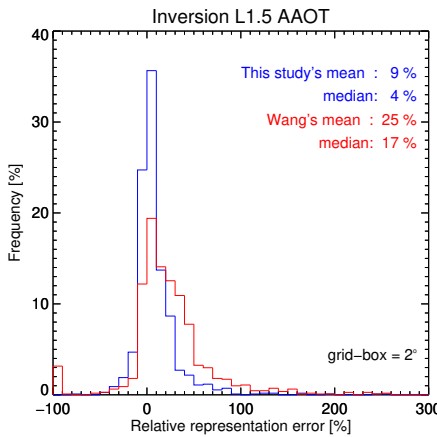

**Figure 14.** Yearly representation errors for AAOT from Inversion L1.5 AERONET as estimated in this paper or using the methodology from Wang et al. (2018) and a model grid-box size of $2^o$. The representation error shown is the spatial representation error (Schutgens et al., 2017), i.e. temporal sampling of observations is ignored.

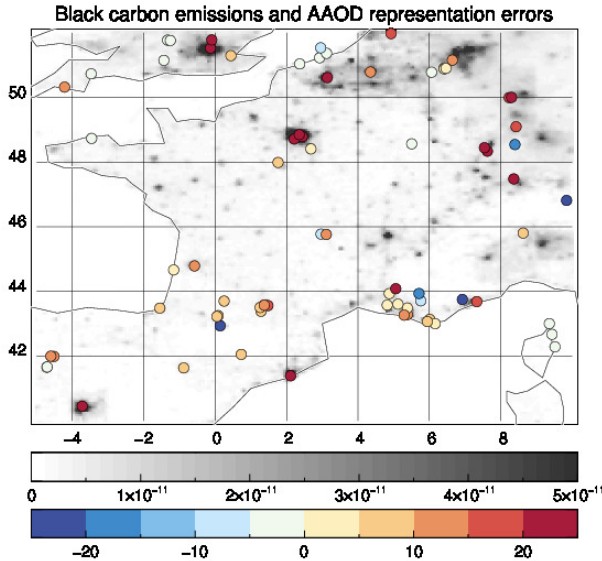

**Figure 13.** Black carbon emissions over France, Europe, with the representation errors in AAOT from Inversion L1.5 AERONET super-imposed. The top colourbar (white-black) represents emissions ([kg/m$^2$s]), and the bottom colourbar (blue-red) represents relative representation errors ([%]). Only *spatial* representation errors are shown, i.e. the temporal sampling of observations is ignored.

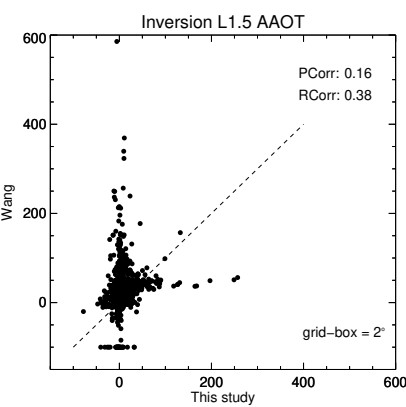

**Figure 15.** Comparison of yearly representation errors for AAOT from Inversion L1.5 AERONET as estimated in this paper or using the methodology from Wang et al. (2018) and a model grid-box size of $2^o$. The representation error shown is the spatial representation error (Schutgens et al., 2017), i.e. temporal sampling of observations is ignored. Also shown are the Pearson linear correlation (PCorr) and rank correlation (RCorr) between the data. The dashed line shows $y = x$.