# Peer review of "Site representativity of AERONET and GAW remotely sensed AOT and AAOT observations"

_Atmospheric Chemistry and Physics, 2019_

## Referee Comment (RC1) · Andrew Sayer (Referee) · 1 Oct 2019

I am writing this review under my own name (Andrew Sayer) as I have a current collaboration with the author. I believe I am able to provide an unbiased review of the paper. I made a few suggestions at the Quick Report stage – the author has addressed these in this version of the study, except for adding a discussion of the paper Li et al (JGR 2013, https://doi.org/10.1002/2016JD025469), which he is still digesting, and a paper of mine still in review (Sayer and Knobelspiesse ACPD 2019, https://www.atmos-chem-phys-discuss.net/acp-2019-372/ ), which will hopefully be accepted in the not-too-distant future.

This paper uses the high-resolution GEOS-5 Nature Run (G5NR) to assess the rep-

resentivity of AERONET and GAW sites for total and absorption aerosol optical thickness (AOT/AAOT) on coarser spatial scales, on various temporal scales and collocation strategies. The sampling of these data sets is applied to G5NR fields, and then compared with averages from the full G5NR. This builds on the author's previous work on representation and sampling-related uncertainties in creating/comparing aerosol data sets, and is a worthwhile extension of sufficient novelty to warrant separate publication.

The analysis is in scope for ACP. The quality of language is pretty good overall. There are a lot of figures but I don't know that it can be condensed much, and the paper is not that too long so I think it's ok. I have a number of comments, below; I'm not really sure whether they fall into minor or major revisions, but I would be interested in reviewing the revised paper.

Page 4 line 7: "sphotometers" - should be sun photometers?

Section 3: This has only one subsection. Could that subheader (3.1) be deleted? Or else another one be added (e.g. for the text summarising the difference between S17 and here)?

Page 7 line 11: Holben (ACP, 2018 https://www.atmos-chem-phys.net/18/655/2018/ ) is a good reference for the DRAGON campaigns, which could be cited here.

Section 4: the evaluation of G5NR is presented mostly in terms of correlation coefficient and regression slope of AERONET vs. G5NR mean and standard deviation of AOT/AAOT. In a sense each site is collapsed down to provide a single data point for the analysis. So this is somewhat different from typical validation analyses where one looks at individual AOT pairings (and in those cases regression is not so appropriate; it is probably fine here, see next paragraph). The reason for this is that G5NR is a nature run so corresponds not to the real (historical observed) world but a realistic world driven by the model. I have used G5NR data before so am familiar with this subtlety, and the author does state it, but I wonder if a less-familiar reader might be confused. I wonder if this point can be hammered-home a bit more with tweaks to working. For example

page 7 line 3 says "simulated AOT shows good agreement with the observations" - this might be changed to read "simulated site-mean AOD" to reinforce the point that we are comparing site averages, not individual points, here. Unless I have misunderstood what is being done. That is one example, but the same applied throughout the section.

More generally the use of correlation and slope can be a bit problematic for AOT analyses, because of the distributions of the data and their error characteristics. It is probably fine here because we are looking at summary statistics for individual sites, rather than individual points themselves, which is a different application from normal. However, because AOD distributions are skewed (and often close to lognormal on timescales like the year evaluated here – see the Sayer and Knobelspiesse reference mentioned above), I wonder if this analysis and Table 5 might be better presented in terms of geometric mean and geometric standard deviation (i.e. in log space). Perhaps the author could do this (doesn't necessarily mean both sets of analysis need to be shown in the paper); if the results are basically the same, great, but if not, it reveals something about limitations of the model simulation.

Page 7 line 20: it might be worth being clearer here that the AERONET AOT requirement for level 2 is 0.40 at 440 nm. For an Ångström exponent (AE) of 2 you get to about 0.25 at 550 nm from this. But for dust-dominated columns with an AE around 0.5 you are around 0.35. So the threshold translates to 550 nm differently dependent on aerosol type. As this threshold is mentioned again on page 9, I think it's worth devoting another line or two to the point here. I realise that the author is using 0.25 as a threshold on the simulation here (i.e. not using the actual thresholds AERONET applies in each case), but that will affect the conclusions systematically at e.g. dust-dominated sites (true AERONET sampling will be poorer than the OSSE suggests because the true AERONET threshold for dust will be more like 0.35 than 0.25).

Page 7 line 21: I think this should be "fewer", not "less" (in both cases), because the observations and sites are countable.

Figure 1: it's not clear what the distinction between solid and dashed lines in the lower panel is here. I know it is pairs of correlation and slope for mean and standard deviation of AOT/AAOT. But I did not see which is which given in the caption or text.

Figure 2: I know there were reviewer and editor comments about number of figures. I think this is one which could potentially be cut (or moved to a supplement) and summarised in the text instead, since the main point (if I understand correctly) is that the statistics for the level 2.0 inversion data are not that different from the less-restrictive level 1.5.

Page 8 line 5-6: I would check in with a member of the AERONET team about this. I don't know what the main uncertainty source leading to AERONET AAOT uncertainties (which are driven by SSA uncertainties is). If it is calibration then that would have an air mass factor dependence so could manifest in apparent daily variation (and violate the author's assumption). If it is something like surface albedo then that may be more of a constant uncertainty which might (consistent with the author's assumption) not affect daily max vs. min AAOT so much. However in Tom Eck's 2014 paper (https://agupubs.onlinelibrary.wiley.com/doi/full/10.1002/jgrd.50500, Figure 4), looking at the variation of SSA at Mongu with day of year, he found different slopes in different years, and attributed this to calibration uncertainties (as the sensor is calibrated before and after each individual deployment, calibration uncertainty is systematic within a year, but random year-to-year). This implies that calibration may be one of the largest contributors, in which case it's possible that the daily variation of SSA (and hence AAOT) is affected (although that paper did not look at SSA diurnal variations). It would probably depend on both the daily variations of SSA and AOT – if AOT varies a lot that may win out over any false signal from SSA. I am not sure whether anyone has looked in great detail but the AERONET team might.

Page 8 lines 12-13: another factor is instrument maintenance issues (e.g. cleaning, replacement when it is sent back for cleaning). Even if this is only 1 week per year then that's still up to 2% coverage (or about 1% when accounting for daylight), which is similar to the difference observed at many sites. So I'd say "meteorological differences and site maintenance issues" or something. This is addressed in the following paragraph but relevant for the direct-Sun data discussed here too.

Page 8 line 17: "several times per day" - I believe it is at specific optical air mass factors but I did a quick look and can't find what those are. I want to say it is a maximum of 6 per day. In the newer data they have hybrid scans nearer solar noon which can extend this, but for the year 2006 simulated by G5NR these were not available. So in that sense the newer AERONET data will fill in some of the gap that is in the observation but not predicted by the model.

Page 8 lines 16-21: One issue is that the inversions require a high degree of azimuthal symmetry (see their QA document at https://aeronet.gsfc.nasa.gov/new_web/Documents/AERONETcriteria_final1.pdf ). So for example if an aerosol plume is thicker to one side of the site than the other, then the scene may be rejected. I don't have a good idea how often this happens; the AERONET team might. I wonder if that is one of the larger factors accounting for the overestimation of AERONET inversion coverage. There are a few other things too, e.g. the AOD threshold for AERONET is stricter for dust-dominated scenes than that applied in this study (see earlier comment) which would affect some of the sites in the tropics.

Page 8 line 25: I believe style guidelines for the journal require sequential appearance of Figures; here Figure 12 is mentioned for the first time, in between 4 and 5. From context it is clear that Figure 12 should not be shifted back here, but the Copernicus style guide disagrees. Perhaps this sentence could be shifted later in the paper instead (and so call back to this section).

Section 5: I realise that this is framed as relative errors throughout. But many applications require absolute uncertainty, so absolute values are also important. So perhaps some text and/or a table could be introduced, with a summary of what fraction

of sites the representation error is smaller than some threshold (perhaps the nominal AERONET AOT uncertainty of 0.01, or the GCOS goal of max[0.03,10%]), for each grid size and time stamp? A large relative sampling uncertainty might be unimportant for a pristine location, for example. Alternatively Figures framed that way could be placed into a Supplement.

Section 5.1, title: I suggest "Representation errors in yearly AOT" to make it clearer up front this is about comparing yearly aggregates colocated in different ways. It will help make the contrast with section 5.2 (monthly) clearer up-front.

Figure 6 caption: "Yeraly" should be "Yearly"

Figures 6, 7 (and dots in 21): can these be regenerated with a different colour bar? The rainbow doesn't print well, emphasises certain parts of the data range but suppresses others, and can't be understood in greyscale or by many colour blind readers. The "viridis" palette is a good alternative, and other options can be found online. Here's a link to an IDL implementation from the CRU: https://crudata.uea.ac.uk/~timo/idl/mkviridis.pro Also, panels are presented as left/right but captions indicate top/bottom, and it would be good to add latitude/longitude labels and/or national borders to this for ease of reference if the reader wants to look up the value for a specific site.

Page 9, lines 4-5: yes, it is clear from this Figure that the bias is negative much more often than it is positive. This implies that higher-AOT times are not sampled by AERONET as often as they should be. One explanation is coincidence (plumes systematically avoid them) but I find that unlikely. So, what is the other mechanism? Could this be the clear-sky bias, i.e. AOT is higher near clouds but near-cloud cases are not sampled? I wonder if there is some way to quickly examine this (e.g. rerun part of the analysis with a cloud fraction threshold of 0.9 instead of 0.01, see if the bias in the representation error shrinks)? Ok, reading ahead to page 10, from Figure 13 it looks like it might be the clear-sky bias. Perhaps that figure and text could be moved up a page. This part –

quantification of clear-sky bias – is to me quite an important result.

Page 9 line 10: this is an important point, I'm glad the author highlighted it again in the Conclusions.

Page 9 line 14: I would say "limitation of" rather than "issue with", to help emphasise this is due to the measurement type rather than being something which was done wrongly.

Page 9 line 22: is -410 m really correct? Which site is 410 m below sea level?

Page 9 lines 29-31: the symbol r was previously used for correlation (e.g. prior paragraph), now is being used for Kinne's rank score. Also, this second use of r does not appear to be stated explicitly in the text. I suggest finding another symbol for the rank score and defining it explicitly in the text. Perhaps capital regular R rather than lower-case italic r.

Page 10 line 1: I would say "typically cannot retrieve aerosol when there are clouds". CALIOP, for example, can retrieve under some clouds. Other retrievals could be extended to do so (see e.g. Lee JGR 2013 https://doi.org/10.1002/jgrd.50806 for an attempt I was be involved with – I don't know that this paper needs to be cited or discussed, just providing it here for an example). I suggest the rephrasing because in part this is a sensor issue but in part it is an algorithm issue.

Page 10 lines 10-11: I am not sure that I follow this. I agree that it will be true if there is correlation from year to year as well. Which there almost certainly is in many parts of the world. But I think that's a bit different from the month-to-month correlations here. I think this should be clarified/spelled out a little more clearly.

Page 10 line 13: I think the words "radiation records" are missing from the end of the Schwarz paper cited here.

Figure 16: what are the dashed lines here?

[Figure]

Page 10 line 21: "criterium" should be "criterion".

Page 10 lines 21-22 and Figures: The impact of the AOT threshold imposed on AAOT representivity is clear. However I am confused because I thought from Table 2, the AOT threshold was taken as 0.03 for level 1.5 data, and not 0.25 (which was for level 2 data). The text (and Figures) here refer to level 1.5 data, but to the 0.25 threshold. Is there a typo here or have I misunderstood? If the threshold was 0.03, why is the bias so positive? If it was 0.25, why are we discussing level 1.5 data and not level 2 data?

Page 11 line 6: there is a missing Figure reference in this line (appears at ??). From context I think that this should be Figure 18, which seems to fit and is not mentioned elsewhere in the paper.

Page 12, lines 11-12: Thank you for making this list available. I downloaded the file from the DOI linked to the citation and it was clear.

Page 13 lines 20-31: I'd personally split this out as a bulleted list (and perhaps the point about the Wang analysis too), to better drawn attention to these conclusions and recommendations.

Figures 8, 9, 10, 12, 13, 14, 15, 17, 18, 19, 20: I think a note should be added here to state that the colours (and, except for Figure 15, numbering legend) follow Figure 5.

As a general question: Is one take away that AERONET and satellites should if possible provide additional hourly products, for intercomparison purposes? Since hourly collocation minimises the representation error for longer-term aggregates, making these more readily available might spur users to use them (rather than the current approach which is more or less monthly collocation).

Language comment: I think in some places the term "uncertainty" should be used instead of "error". The calculation of representation error via difference between the differently-sampled G5NR simulation is an error. But I think when talking in a larger sense, we are using this representation error (from the OSSE) to estimate the actual

representation uncertainty (which we don't know for sure). Also when talking about AERONET inversions, we should be typically talking about the uncertainty in the retrieval (as the error is not known). I suggest checking individual uses of these terms in the papers.

---

## Referee Comment (RC2) · Anonymous Referee #4 · 11 Oct 2019

The author uses a two year (June 2005 to May 2007), atmosphere-only (prescribed sea surface temperatures), free-running (not nudged), global high-resolution (0.0625 degree) simulation (GEOS5 Nature Run) to conduct an Observing System Simulation Experiment (OSSE) with which the representativeness of aerosol observations (aerosol optical thickness, AOT, and absorbing aerosol optical thickness, AAOT) at AERONET and GAW sites is assessed with regard to space (from 0.0625 to 4.0 degrees) and time (from hourly to annual means).

AOT and AAOT are an important facet of historical and current climate. Their modeling remains challenging, as does comparison of simulation results and observations (from in-situ to satellite). The paper deals with an associated, critical aspect: the spatial and temporal representativeness of in-situ observations of AOT and AAOT on a global

scale. Specifically, monthly representation errors are found to be correlated, implying that annual means offer only a moderate gain in representativeness. Moreover, the global representation bias of AAOT (relevant for model evaluation) is estimated below 10%, showing at the same time that the corresponding 30% estimate by Wang et al. (2018) can be attributed to too coarse model data used in that study.

Recent years have seen a few studies dealing with the topic of representativeness in general and for aerosol observations in particular, yet consensus has not yet been reached. The present paper with its quantitative, objective, and global perspective is a substantial, high-quality contribution to the discussion. I recommend publication in ACP after some revisions.

**Specific comments:**

page 3, line 9: It would be good to add a bit more information on the simulation data, notably that it is a free running (not nudged) simulation, possibly also a word on vertical resolution and output frequency (hourly or even less?; how 'high-resolution' is the model data with regard to time?).

page 3, line 24: Replace AOD with AOT, here and throughout the manuscript; likewise for AAOD and AAOT.

page 4, line 8: What do you mean by "here we will assume a potentially remotely sensed columnar product ... and consider its representation errors"?

page 4, line 11: Given that various definitions of "representation error" exist in the literature, it would be helpful if the author could provide the exact definition he uses in this paper (e.g. reference to another paper; formula; description).

page 4, line 14: Here it is said that this work deals mostly with yearly and some monthly averages, yet many figures show hourly data. Please clarify.

page 6, line 19: What do you mean by the sing-less error? Absolute error or rootmean-square?

page 7, line 4: I assume that by 'correlation' you mean R, not $R^2$. It may be helpful for the reader to explicitly say so.

page 8, line 22: When it is said that G5NR seems capable to realistically simulate the spatial variation of AOT and AAOT, "spatial" here seems to refer to different sites. It is not shown, it seems to me, how realistically G5NR captures the spatial variability of AOT and AAOT around a single site, including the adopted averaging distances between 0.5 and 4 degrees. It may be worthwhile to clarify this point.

page 8, line 31: As grid box sizes are reduced, hourly collocation errors are reduced. Could this be because the physical connection (same cause, exchange of signal) between two hourly time series at two distant points decreases with distance? Could the author comment on why the reported finding is (or is not) physically plausible?

page 9, line 5: Can something be said as to the (physical?) causes of the found east-west (North America) and north-south (Europe) gradient in representativeness?

page 9, line 21: Apart from the shorter atmospheric column, could it also matter that high lying mountain sites are often in the 'free troposphere', i.e., (somewhat) decoupled from the sources of (short lived) aerosols in the boundary layer?

page 11, line 9: Does it matter here, how missing values are treated when computing the annual mean?

page 12, line 23: The author mentions once more the calculated meteorology. Overall, he seems to claim / find that meteorology is not that important for representativeness. Is this indeed what he means to say? And, if so, how about phenomena like ENSO? Could, for example, the comparatively bad performance of South America be related to the presence / absence of ENSO in the model data?

page 13, lines 13 and 20: Does this imply that meteorology is not that important for representativeness?

page 13, line 24: It is not clear where the error of typically 20% globally comes from, I do not see this in the main text of the paper.

Figure 1: One may add in the caption what the different line-styles in the lower row mean.

Figure 5: Any idea why there is an overall bias towards negative values? It seems unlikely that the (few) high lying GAW sites (and their shorter atmospheric column) alone can serve as an explanation.

Figure 6: Any idea what the (physical?) reason is behind the found spatial gradients?

Figure 8: Any idea why Europe is so good and South America rather bad? Geography? ENSO? Number of sites? Other?

Figure 12: Maybe refer in the caption to table 6 (explanation of r). Also, the figure seems to suggest that there is no connection between "r" from Kinne et al. and the relative representation error from this paper; the bars in the plot look pretty much the same for "all", "r=0", "r=1", and even "r>1" for yearly data. Please comment.

Figure 16: What are the dashed lines?

**Specific comments:**

page 1, line 16: due *to* methodological choices

page 2, line 14: remove S16b

page 10, line 20: "for for" should read "for"

page 11, line 6: "Fig.??" should be properly referenced

---

## Referee Comment (RC3) · Anonymous Referee #3 · 11 Oct 2019

**Summary**

In this well written manuscript the author conducts an observing system simulation experiment (OSSE) to study the representativity of remotely sensed aerosol optical thickness (AOT) and absorbing aerosol optical thickness (AAOT) observations made at the locations of AERONET and GAW stations. For the OSSE, GEOS-5 Nature Run (G5NR) simulation data for year 2016 is used. The scientific methods and assumptions in the manuscript are valid and mainly clearly outlined. The experiment is mostly described with sufficient details to make reproduction of the results possible. The author evaluates the OSSE and G5NR with real AERONET data and shows that at least for AOT the OSSE and G5NR have significant skill and the conclusions may therefore be generalised to real observations. The manuscript explores the representativity of AOT

and AAOT in many different ways and discusses the findings with sufficient details. The conclusions given are supported by the results given in the manuscript. The manuscript is an extension to the previous work carried out by the author and it contains new contributions to the field of aerosol remote sensing and modeling. The manuscript will be a valuable resource for everyone carrying out AOT comparisons between models and observations and I recommend that the manuscript should be published in ACP after minor revision.

**General Comments**

There are 23 figures in the manuscript. Many of the figures have very short and incomplete captions and the reader is required to find another figure with more complete caption to fully understand the figures. Also in Figure 7 the caption refers to a colour bar that is presented in other figure. The author should make the figures more self-explanatory for example by adding missing colour bar and add more details to figure captions.

Sometimes figures are referred by using "Fig." and sometimes "Fig". Please see the journal instructions and correct.

Throughout the manuscript, the representation erros are shown as relative representation error. It would be interesting to see or at least have a comment on the absolute representation errors. This would show if high representation errors mainly correspond to small AOT values only or are there relatively large errors present also in cases with large AOT.

**Specific Comments**

p.2 l.14 "return times" Would "overpass times" or "revisit times" be more commonly used term to be used here?

p.5 l.1 "The maximum cloud-fraction was slightly tuned..." Please clarify what you mean by "slight tuning".

p.6 l.7 "...we will limit our analysis to latitudes below 60°." In Figure 4, there are stations at above 60 degrees.

p.7 l.9 For reproduction of the results, please list the sites that were removed from the analysis.

p.7 l.20 Is AOT threshold of 0.25 correct? To my understanding the threshold at 440 nm is 0.4 and it depends on the spectral dependence of AOT (Angstrom Exponent) what it will be at 550 nm. So for me this seems a bit low value for the threshold. Please make sure the reader understand that you have used a "non-standard" value of 0.25 or correct to match the true AERONET threshold (throughout the manuscript, same limit mentioned for example on p.9 l.13).

p.9 l.22 Altitude of -410 meters, is this correct?

p.9 l.29 Here notation "r" is used for representation ranking by Kinne et al. (2013). In some parts of the manuscript "r" is used to denote correlation coefficient so there is a conflict here. Please correct throughout the manuscript to remove the possible misunderstandings.

p.13 l.12 "G5NR and the OSSE are evaluated and found to show significant skill." This result was found for AOT, not for AAOT. Please clarify that this statement applies only to AOT to avoid misunderstandings.

p.18 Figure 1 Bottom row, what are the differences between solid and dashed lines?

p.21 Figure 9 Please define DS. Also on the upper right corner the text is overlapping with the figure and may be difficult to read.

p.25 Figure 16 What are the dashed lines?

p.27 Figure 20 "r" is not defined.

p.28 Figure 21 If possible, please add the another colour bar from Fig. 7.

[Figure]

**Typos**

p.2 l.5 "As aerosol are..." missing "s"

p.2 l.14 "S16b"?

p.8 l.20 "nd"

p.9 l.34 "Sofar"

p.10 l.1 "...can not observed..."

p.10 l.20 "Sgnificant"

p.11 l.6 "Fig. ??."

---

## Referee Comment (RC4) · Anonymous Referee #2 · 17 Nov 2019

This paper presents results assessing the representation errors for observations of AERONET AOT and AAOT, and GAW AOT (and surface absorption aerosol), using a modeled GEOS5 Nature Run and an OSSE, for hourly, daily, monthly, and yearly protocols. A particularly interesting result is the comparison to results from Wang et al., 2018, with the finding that model downscaling is responsible for the larger biases in the former paper compared with the current one. I would recommend publication provided the following comments are addressed.

Major comments:

There are two places in the paper where it appears representativity/bias-based conclusions are drawn based on information that could be mostly influenced by aerosol type:

[Figure]

1) P. 7, line ~15: "As the minimum number of observations per site increases, so do the correlations, probably due to a reduction in statistical noise (partly due to different simulated and actual meteorologies). But the overall bias also increases." The author then goes on to state that the sites with very high minima of observations are dust-dominated, and dust AOT is overestimated.

2) P. 11, line ~8: "for yearly collocation protocol which allows significant biases for sites in South America and Africa. This is related to the AOT criteri[on] for valid observations and the dominant influence of episodic biomass burning for these two continents"

These seem to be potentially interesting results which are not discussed in this paper: representation errors also vary based on the aerosol type itself (specifically for the dust case). What happens if the dust sites are explicitly excluded? Does the correlation/noise/bias trend remain? Can you disentangle those two effects? What's the non-dust site with the highest number of observations? Since South America/Africa are such outliers, is there some lesson to be had from how seasonal sampling variability affects these errors?

This paper has a lot of figures which show generally similar results (trends from yearly to daily collocation protocols for different subsets are fairly similar) but have somewhat limited discussion of each. I wonder if it would be possible to consolidate a few of them to more clearly distill the main story of the paper, but it's probably fine if everything stays in. I would recommend more detailed figure captions, though; as-is the individual figures are a bit hard to follow. Also, does Fig 17 have a different collocation protocol for the brown bars than the others?

p. 12, Line 11: I might clarify that this ranking at the DOI under Schutgens (2019) is the same as that which was mentioned p. 9, Line 33. It would be interesting to have a comment as to how this has changed since the Kinne et al., 2013 rankings, since if I understand correctly this new ranking has up to a full decade of additional data?

Minor comments:

[Figure]

-p. 3 , Lines 28-29: I believe CERES "cloud fractions" are derived from their collocated MODIS instruments.

-p. 7 Line 5: "correlation ($\sim$0.45)". To what does this $\sim$ refer?

-p. 7 Line 20: as the other reviewers said, this is not strictly true; the L2.0 data have a minimum AOT of 0.4 at 440nm, which here has been interpolated to $\sim$0.25 at 550nm. I'd clarify this point.

-Figure 1: which is the solid and which is the dashed line? This should have a caption.

-Figure 21: the color bar from Fig 7 should be reproduced here; also there should be units added to the BC emissions shading.

-throughout the paper, I believe the singular form of "criteria" should be "criterion," not "criterium."

-Figures 6 and 7: captions say top/bottom, but should say left/right. Also "yearly"

-Figure 23: this figure could benefit from a 1:1 line to guide the eye.

-I also found many typos, misspellings, and minor grammatical errors throughout, some of which were mentioned by the other reviewers. I'd recommend a careful readthrough for such minor errors before publication. Some of them are below.

p. 1 Line 12: "is advocated instead"

p. 2 Line 14: "S16b In this paper" ?

p. 4 Line 7 "sun photometers"; Line 17 "sub-sampling"

Table 5: "southern"

p. 7 Line 14: "observations"

p. 8 Line 3: comma after Again; Line 4: "its"; Line 20 "Sahara and Saudi Arabia"

p. 9 Line 29: extra )

p. 10 Line 1: "cannot observe".

p. 10 Line 20: "significant"

p. 11 Line 6: broken reference to Fig [I assume 18].

p. 11 Line 9: "many fewer observations are made"
* * *

---

## Author Response (AR1)

**Response to all reviewers**

Before detailing my specific response to each reviewer, a few general comments:

- Figures that contain maps have now been remade with colour schemes that are hopefully more readable to colour blind people.
- Several figures have been moved to a supplement (mostly figures to do with the AAOT analysis)
- A few figures have been removed as I considered them too technical for a general audience (e.g. the figure used to discuss the nature of representation errors per site: random vs bias)
- Figure 1 (evaluation of G5NR) has been remade as I realized that I plotted Root Mean Square Differences instead of standard deviations (in blue). This partially changes the figures (results for the site mean are unaffected). Overall conclusions do not change much, except the poor performance of G5NR in temporal variability of AAOT is more pronounced.
- A table with absolute values of representation errors has been added, to compare against measurement error (representation errors are significantly larger).

**Response to reviewer 1 (Andy Sayer)**

I'd like to thank Andy for his time and many useful comments. I think the paper has improved in clarity as a result. The on-going discussion on how to calculate annual averages (arithmetic vs geometric) is also an interesting one, and I'm happy to contribute.

The reviewer suggests condensing the paper. Other reviewers have suggested this as well, pointing out the use of supplementary pages. I have decided to move part of the AOT representation discussion (e.g. variations by regions) and the entire AAOT representation discussion to a supplement. That should significantly shorten the main paper, without detracting from the main conclusions. The original AAOT analysis will be available for those with an interest in it.

Page 4 line 7: "sphotometers" – should be sun photometers?

Corrected.

Section 3: This has only one subsection. Could that subheader (3.1) be deleted? Or else another one be added (e.g. for the text summarising the difference between S17 and here)?

Deleted.

Page 7 line 11: Holben (ACP, 2018 https://www.atmos-chem-phys.net/18/655/2018/ ) is a good reference for the DRAGON campaigns, which could be cited here.

Agreed

Section 4: the evaluation of G5NR is presented mostly in terms of correlation coeffi- cient and regression slope of AERONET vs. G5NR mean and standard deviation of AOT/AAOT. In a sense each site is collapsed down to provide a single data point for the analysis. So this is somewhat different from typical validation analyses where one looks at individual AOT pairings (and in those cases regression is not so appropriate; it is probably fine here,

see next paragraph). The reason for this is that G5NR is a nature run so corresponds not to the real (historical observed) world but a realistic world driven by the model. I have used G5NR data before so am familiar with this subtlety, and the author does state it, but I wonder if a less-familiar reader might be confused. I wonder if this point can be hammered-home a bit more with tweaks to working. For example page 7 line 3 says "simulated AOT shows good agreement with the observations" – this might be changed to read "simulated site-mean AOD" to reinforce the point that we are comparing site averages, not individual points, here. Unless I have misunderstood what is being done. That is one example, but the same applied throughout the section.

I agree that I can do more to impress upon the reader this is a free run. Very interestingly, yearly AOT per site agrees reasonably well with observations. While in satellite research, it is more common to provide error statistics on daily scales, in model research longer time-scales are more usual. First of all, we want to be able to represent the "base" state of the atmosphere (I do provide additional information in the standard deviation, i.e. variability per site, of AOT). The correlation in these yearly values expresses the ability of G5NR to realistically simulate the spatial distribution of annual AOT (at scales of AERONET separation distances).

More generally the use of correlation and slope can be a bit problematic for AOT analy- ses, because of the distributions of the data and their error characteristics. It is probably fine here because we are looking at summary statistics for individual sites, rather than individual points themselves, which is a different application from normal. However, because AOD distributions are skewed (and often close to lognormal on timescales like the year evaluated here — see the Sayer and Knobelspiesse reference mentioned above), I wonder if this analysis and Table 5 might be better presented in terms of geo- metric mean and geometric standard deviation (i.e. in log space). Perhaps the author could do this (doesn't necessarily mean both sets of analysis need to be shown in the paper); if the results are basically the same, great, but if not, it reveals something about limitations of the model simulation.

An interesting idea and easily implemented. I have followed Sayer & Knobelspiesse with interest and suspect we will have many discussions on such issues in upcoming AEROCOM/AEROSAT meetings!

Below I show the evaluation of G5NR, using either arithmetic (as in my paper) or geometric (as advocated by Sayer & Knobelspiesse) means. For definition of geometric mean and standard deviation:
https://en.wikipedia.org/wiki/Geometric_mean and
https://en.wikipedia.org/wiki/Geometric_standard_deviation

[Figure]

AERONET DirectSun L2.0 (arithmetic): AOT

Mean
Mean:      0.00
StdDev:    0.12
PCorr:     0.75
OLS s:     0.81
Nr:        216

StdDev
Mean:     −0.00
StdDev:    0.11
PCorr:     0.61
OLS s:     0.50
Nr:        216

[Figure]

AERONET DirectSun L2.0 (geometric): AOT

Mean
Mean:      0.00
StdDev:    0.10
PCorr:     0.76
OLS s:     0.95
Nr:        216

StdDev
Mean:      0.01
StdDev:    0.35
PCorr:     0.44
OLS s:     0.46
Nr:        216

Using geometric mean and standard deviation has the following consequences for network statistics (the text in the figure).

- Bias in mean AOT per site hardly changes
- Spread in mean AOT per site decreases by 20%. However, mean AOT per site also decreases by about 20%, so this is not surprising.
- Correlation for mean AOT per site hardly changes
- Regression slope for mean AOT per site improves significantly
- Standard deviation AOT per site now shows rather large values and significantly lower correlation.

If I calculate standard deviation AOT per site from an arithmetic mean over logarithmic AOT (as we discussed off-line), evaluation is still poorer than when using arithmetic mean over AOT.

In short, I see no significant improvements in evaluation statistics when using a geometric mean. The exception would be the regression slope and I think it is worthwhile to explore this further. The use of geometric standard deviation has a negative impact on the correlation and should be used with caution.

Page 7 line 20: it might be worth being clearer here that the AERONET AOT require- ment for level 2 is 0.40 at 440 nm. For an Ångström exponent (AE) of 2 you get to about 0.25 at 550 nm from this. But for dust-dominated columns with an AE around 0.5 you are around 0.35. So the threshold translates to 550 nm differently dependent on aerosol type. As this threshold is mentioned again on page 9, I think it's worth devoting another line or two to the point here. I realise that the author is using 0.25 as a thresh- old on the simulation here (i.e. not using the actual thresholds AERONET applies in each case), but that will affect the conclusions systematically at e.g. dust-dominated sites (true AERONET sampling will be poorer than the OSSE suggests because the true AERONET threshold for dust will be more like 0.35 than 0.25).

I agree. This was also pointed out by another reviewer. As you say yourself, it essentially means that over dusty sites I present a best case for the representation errors in Inversion L2.0. More importantly, though, is that the brunt of my analysis concerns Inversion *L1.5* and this will not be affected by the threshold.

Page 7 line 21: I think this should be "fewer", not "less" (in both cases), because the observations and sites are countable.

Corrected. I thought it sounded strange but couldn't pinpoint why ☺.

Figure 1: it's not clear what the distinction between solid and dashed lines in the lower panel is here. I know it is pairs of correlation and slope for mean and standard deviation of AOT/AAOT. But I did not see which is which given in the caption or text.

This has been corrected in the caption.

Figure 2: I know there were reviewer and editor comments about number of figures. I think this is one which could potentially be cut (or moved to a supplement) and sum- marised in the text instead, since the main point (if I understand correctly) is that the statistics for the level 2.0 inversion data are not that different from the less-restrictive level 1.5.

Correct. I also feel several figures can be moved to a supplement, Fig. 2 included.

Page 8 line 5-6: I would check in with a member of the AERONET team about this. I don't know what the main uncertainty source leading to AERONET AAOT uncertain- ties (which are driven by SSA uncertainties is). If it is calibration then that would have an air mass factor dependence so could manifest in apparent daily variation (and vi- olate the author's assumption). If it is something like surface albedo then that may be more of a constant uncertainty which might (consistent with the author's assump- tion) not affect daily max vs. min AAOT so much. However in Tom Eck's 2014 paper (https://agupubs.onlinelibrary.wiley.com/doi/full/10.1002/jgrd.50500, Figure 4), looking at the variation of SSA at Mongu with day of year, he found different slopes in different years, and attributed this to calibration uncertainties (as the sensor is calibrated be- fore and after each individual deployment, calibration uncertainty is systematic within a year, but random year-to-year). This implies that calibration may be one of the largest contributors, in which case it's possible that the daily variation of SSA (and hence AAOT) is affected (although that paper did not look at SSA diurnal variations). It would probably depend on both the daily variations of SSA and AOT — if AOT varies a lot that may win out over any false signal from SSA. I am not sure whether anyone has looked in great detail but the AERONET team might.

I appreciate your points and I may have been too positive about this. But my line of thinking is that the differencing inherent in a daily MAX — MIN AAOT value mitigates the impact of retrieval errors.

If such errors are constant throughout the day, this is a trivial statement.

If such errors behave entirely random, a yearly average of the difference will not be affected much either

Obviously, correlated but time-varying errors do exist and can be introduced by e.g. a calibration error. But in that case, the error in the difference is unlikely to be larger than the error in individual AAOT.

I've contacted Tom Eck (AERONET) and Oleg Dubovik (Lille U., developer of original Inversion scheme) about this issue and they agreed with my reasoning. Proper research is probably needed to put this on a firmer footing and I'll amend the text accordingly.

Page 8 lines 12-13: another factor is instrument maintenance issues (e.g. cleaning, replacement when it is sent back for cleaning). Even if this is only 1 week per year then that's still up to 2% coverage (or about 1% when accounting for daylight), which is simi- lar to the difference observed at many sites. So I'd say "meteorological differences and site maintenance issues" or something. This is addressed in the following paragraph but relevant for the direct-Sun data discussed here too.

Agreed.

Page 8 line 17: "several times per day" – I believe it is at specific optical air mass factors but I did a quick look and can't find what those are. I want to say it is a maximum of 6 per day. In the newer data they have hybrid scans nearer solar noon which can extend this, but for the year 2006 simulated by G5NR these were not available. So in that sense the newer AERONET data will fill in some of the gap that is in the observation but not predicted by the model.

I was deliberately vague as I don't think this is relevant at this stage. The OSSE overestimates observational coverage, and this can be due to a whole list of reasons. Note that my sensitivity study suggests that this over-estimation of temporal coverage (e.g. Fig 9 and ) has no large impact.

Page 8 lines 16-21: One issue is that the inversions require a high degree of azimuthal symmetry (see their QA document at https://aeronet.gsfc.nasa.gov/new_web/Documents/AERONETcriteria_final1.pdf ). So for example if an aerosol plume is thicker to one side of the site than the other, then the scene may be rejected. I don't have a good idea how often this happens; the AERONET team might. I wonder if that is one of the larger factors accounting for the overestimation of AERONET inversion coverage. There are a few other things too, e.g. the AOD threshold for AERONET is stricter for dust-dominated scenes than that applied in this study (see earlier comment) which would affect some of the sites in the tropics.

I am aware of these issues. As a matter of fact, because it requires high degrees of azimuthal symmetry, Inversion has a built-in check for *spatial* representation errors. That check will in turn lower temporal coverage at any site. I had no idea how to represent that so left it out of the OSSE. But it is worth discussing this.

Page 8 line 25: I believe style guidelines for the journal require sequential appearance of Figures; here Figure 12 is mentioned for the first time, in between 4 and 5. From context it is clear that Figure 12 should not be shifted back here, but the Copernicus style guide disagrees. Perhaps this sentence could be shifted later in the paper instead (and so call back to this section).

I prefer to leave it as it is. It is sometimes unavoidable that one figure is referenced several times in a paper. I have used the location of the main discussion of a figure in the text to order the figures.

Section 5: I realise that this is framed as relative errors throughout. But many appli- cations require absolute uncertainty, so absolute values are also important. So per- haps some text and/or a table could be introduced, with a summary of what fraction of sites the representation error is smaller than some threshold (perhaps the nominal AERONET AOT uncertainty of 0.01, or the GCOS goal of max[0.03,10%]), for each grid size and time stamp? A large relative sampling uncertainty might be unimportant for a pristine location, for example. Alternatively Figures framed that way could be placed into a Supplement.

Agreed.

Section 5.1, title: I suggest "Representation errors in yearly AOT" to make it clearer up front this is about comparing yearly aggregates colocated in different ways. It will help make the contrast with section 5.2 (monthly) clearer up-front.

Agreed.

Figure 6 caption: "Yeraly" should be "Yearly"

Corrected.

Figures 6, 7 (and dots in 21): can these be regenerated with a different colour bar? The rainbow doesn't print well, emphasises certain parts of the data range but suppresses others, and can't be understood in greyscale or by many colour blind readers. The "viridis" palette is a good alternative, and

other op- tions can be found online. Here's a link to an IDL implementation from the CRU: https://crudata.uea.ac.uk/~timo/idl/mkviridis.pro Also, panels are presented as left/right but captions indicate top/bottom, and it would be good to add latitude/longitude labels and/or national borders to this for ease of reference if the reader wants to look up the value for a specific site.

Thanks for the link. Figures will be remade.

Page 9, lines 4–5: yes, it is clear from this Figure that the bias is negative much more of- ten than it is positive. This implies that higher-AOT times are not sampled by AERONET as often as they should be. One explanation is coincidence (plumes systematically avoid them) but I find that unlikely. So, what is the other mechanism? Could this be the clear-sky bias, i.e. AOT is higher near clouds but near-cloud cases are not sampled? I wonder if there is some way to quickly examine this (e.g. rerun part of the analysis with a cloud fraction threshold of 0.9 instead of 0.01, see if the bias in the representation error shrinks)? Ok, reading ahead to page 10, from Figure 13 it looks like it might be the clear-sky bias. Perhaps that figure and text could be moved up a page. This part — quantification of clear-sky bias — is to me quite an important result.

It is the clear-sky bias. My code generates error estimates for individual masking factors (daytime/nighttime, cloudiness, lower AOT threshold) and identification of the main cause is easy. I will move this discussion forward.

Page 9 line 10: this is an important point, I'm glad the author highlighted it again in the Conclusions.

I think it may similarly have consequences for AEROCOM model evaluations

Page 9 line 14: I would say "limitation of" rather than "issue with", to help emphasise this is due to the measurement type rather than being something which was done wrongly.

Agreed.

Page 9 line 22: is –410 m really correct? Which site is 410 m below sea level?

Dead_Sea

Page 9 lines 29–31: the symbol r was previously used for correlation (e.g. prior para- graph), now is being used for Kinne's rank score. Also, this second use of r does not appear to be stated explicitly in the text. I suggest finding another symbol for the rank score and defining it explicitly in the text. Perhaps capital regular R rather than lower-case italic r.

Kinne uses "r" so I'd like to use it as well. But I will make sure it's clear this is a different "r" from the rest of the paper.

Page 10 line 1: I would say "typically cannot retrieve aerosol when there are clouds". CALIOP, for example, can retrieve under some clouds. Other retrievals could be ex- tended to do so (see e.g. Lee JGR 2013 https://doi.org/10.1002/jgrd.50806 for an attempt I was be involved with — I don't know that this paper needs to be cited or dis- cussed, just providing it here for an example). I suggest the rephrasing because in part this is a sensor issue but in part it is an algorithm issue.

Ok.

Page 10 lines 10-11: I am not sure that I follow this. I agree that it will be true if there is correlation from year to year as well. Which there almost certainly is in many parts of the world. But I think that's a bit different from the month-to-month correlations here. I think this should be clarified/spelled out a little more clearly.

Note that I am talking about the increase of correlation in 2006 between January and months like November and December (11 or 12 months apart!). Obviously, I can't prove this is repeated every year but there are good reasons to assume this will happen.

Page 10 line 13: I think the words "radiation records" are missing from the end of the Schwarz paper cited here.

Thanks.

Figure 16: what are the dashed lines here?

Y=2x and x=2y, for convenience. Now explained in caption.

Page 10 line 21: "criterium" should be "criterion".

Corrected.

Page 10 lines 21-22 and Figures: The impact of the AOT threshold imposed on AAOT representivity is clear. However I am confused because I thought from Table 2, the AOT threshold was taken as 0.03 for level 1.5 data, and not 0.25 (which was for level 2 data). The text (and Figures) here refer to level 1.5 data, but to the 0.25 threshold. Is there a typo here or have I misunderstood? If the threshold was 0.03, why is the bias so positive? If it was 0.25, why are we discussing level 1.5 data and not level 2 data?

Thanks. It would appear that an earlier edit went wrong. Clearly, the AOT>0.25 statement has no relevance here.

Page 11 line 6: there is a missing Figure reference in this line (appears at ??). From context I think that this should be Figure 18, which seems to fit and is not mentioned elsewhere in the paper.

Corrected.

Page 12, lines 11-12: Thank you for making this list available. I downloaded the file from the DOI linked to the citation and it was clear.

You're welcome. Comments always welcome, also after publication. This will hopefully be an evolving document.

Page 13 lines 20-31: I'd personally split this out as a bulleted list (and perhaps the point about the Wang analysis too), to better drawn attention to these conclusions and recommendations.

Thanks for the suggestion.

Figures 8, 9, 10, 12, 13, 14, 15, 17, 18, 19, 20: I think a note should be added here to state that the colours (and, except for Figure 15, numbering legend) follow Figure 5.

Ok.

As a general question: Is one take away that AERONET and satellites should if possible provide additional hourly products, for intercomparison purposes? Since hourly collo- cation minimises the representation error for longer-term aggregates, making these more readily available might spur users to use them (rather than the current approach which is more or less monthly collocation).

But don't these data already come at hourly or daily resolution? Of course, users seem fond of the monthly L3 products and I am not sure how to change that. Removing monthly L3 data from archives would be my preferred option but I can see that would be unpopular.

Language comment: I think in some places the term "uncertainty" should be used instead of "error". The calculation of representation error via difference between the differently-sampled G5NR simulation is an error. But I think when talking in a larger sense, we are using this representation error (from the OSSE) to estimate the actual representation uncertainty (which we don't know for sure). Also when talking about AERONET inversions, we should be typically talking about the uncertainty in the re- trieval (as the error is not known). I suggest checking individual uses of these terms in the papers.

Ok.

**Response to reviewer 2**

I'd like to thank the reviewer for their time and many useful comments. I think the paper has improved as a result of their feedback.

The reviewer suggests condensing the paper. Other reviewers have suggested this as well, pointing out the use of supplementary pages. I have decided to move part of the AOT representation discussion (e.g. variations by regions) and most of the AAOT representation discussion to a supplement. That should significantly shorten the main paper, without detracting from the main conclusions. The original AAOT analysis will be available for those with an interest in it.

[.. ] These seem to be potentially interesting results which are not discussed in this pa- per: representation errors also vary based on the aerosol type itself [..]

I think there is definitely room for a study on the impact of aerosol species on representation errors. Here and there in my papers I have alluded to this. E.g. in S16a the differences between black carbon, sea salt and sulfate were briefly discussed. It is not so much the species itself but the spatio-temporal distribution of its sources that is the important factor. A proper investigation would be outside the scope of the present paper which already is quite large. For the impact of dust on the G5NR evaluation (which is different from an analysis of representation errros), see e.g. Table 5.

Does Fig 17 have a different collocation protocol for the brown bars than the others?

No, it doesn't but I forgot to update the caption to bring it in line with other figures. Changed now.

p. 3 , Lines 28-29: I believe CERES "cloud fractions" are derived from their collocated MODIS instruments.

The reviewer is correct but I prefer to stick with CERES cloud fraction as this is the form used in Gelaro et al. 2015. They are cloud fractions derived from MODIS, specifically for CERES.

p. 7 Line 5: "correlation (~0.45)". To what does this ~ refer?

It means, "about". I believe this is standard usage: https://en.wikipedia.org/wiki/Tilde#Mathematics .

p. 7 Line 20: as the other reviewers said, this is not strictly true; the L2.0 data have a minimum AOT of 0.4 at 440nm, which here has been interpolated to ~0.25 at 550nm. I'd clarify this point.

The reviewers are of course correct and this point will be clarified.

-Figure 1: which is the solid and which is the dashed line? This should have a caption.

Thank you. Red solid is correlation, red dashed is slope, blue solid is mean, blue dashed is standard deviation. This information has been added to the caption.

Figure 21: the color bar from Fig 7 should be reproduced here; also there should be units added to the BC emissions shading.

I'll try to add a colour bar for the representation errors. The unit for bc emissions is mentioned already in the title.

-throughout the paper, I believe the singular form of "criteria" should be "criterion," not "criterium."

Changed.

-Figures 6 and 7: captions say top/bottom, but should say left/right. Also "yearly"

Changed. Actually, the top/bottom issue is due to different formats used for Discussions and Final publications.

-Figure 23: this figure could benefit from a 1:1 line to guide the eye.

Ok.

And thanks for bringing those typos and misspellings to my attention. They have been corrected.

**Response to reviewer 3**

I'd like to thank the reviewer for their time and many useful comments. I think the paper has improved in clarity as a result of their feedback.

The reviewer suggests condensing the paper. Other reviewers have suggested this as well, pointing out the use of supplementary pages. I have decided to move part of the AOT representation discussion (e.g. variations by regions) and most of the AAOT representation discussion to a supplement. That should significantly shorten the main paper, without detracting from the main conclusions. The original AAOT analysis will be available for those with an interest in it.

Sometimes figures are referred by using "Fig." and sometimes "Fig".

All changed to "Fig.".

It would be interesting to see or at least have a comment on the absolute representation errors. This would show if high representation errors mainly correspond to small AOT values only or are there relatively large errors present also in cases with large AOT.

A good point. I started using relative errors in S16b as it allows more easily a comparison across different types of measurement. Also in the current paper, it allows comparison of AOT and AAOT representation errors. But I will include a paragraph on absolute values of these errors.

p.2 l.14 "return times" Would "overpass times" or "revisit times" be more commonly used term to be used here?

Yes, I'll use revisit times.

p.5 l.1 "The maximum cloud-fraction was slightly tuned..." Please clarify what you mean by "slight tuning".

Yes, I can see how this is confusing. I can choose values between 0 and 1 and ended up using 0.01 because the results agree *slightly* better with the observations. As the original text stated, the impact is small. Also, I only explored five different values (0.01, 0.1, 0.5, 0.9, 0.99) so in that sense the tuning was coarse. I have removed 'slight'.

p.6 l.7 "...we will limit our analysis to latitudes below 60$^{\circ}$." In Figure 4, there are stations at above 60 degrees.

True, AERONET sites exist at higher latitudes and my data included those as well. For the evaluation of G5NR, I used all sites. For the representation study, I included only sites below 60 latitude (except in Fig 4 & 5). Text now reflects this.

p.7 l.9 For reproduction of the results, please list the sites that were removed from the analysis.

They were only removed from the analysis for one result (Table 5, line Europe*). Throughout the paper they have been used in the analysis of representation errors. I have amended the text to clarify this.

p.7 l.20 Is AOT threshold of 0.25 correct? To my understanding the threshold at 440 nm is 0.4 and it depends on the spectral dependence of AOT (Angstrom Exponent) what it will be at 550 nm. So for me this seems a bit low value for the threshold. Please make sure the reader understand that you have used a "non-standard" value of 0.25 or correct to match the true AERONET threshold (throughout the manuscript, same limit mentioned for example on p.9 l.13).

The reviewer is correct. The text will be modified accordingly. Note that this has almost no impact on the paper as I mostly study Inversion L1.5 data. Only in Fig 9, where a comparison is made between L1.5 and L2.0 will this affect the L2.0 analysis (i.e. representation errors will be underestimated). Note that this issue (AOD@550nm >0.25 instead of AOD@440nm >0.4) will mostly affect dusty stations (for which AOD@550 ~ AOD@440). Since most of my statistics are based on year-averages from stations and dusty stations form a minority, I do not expect very big changes.

p.9 l.22 Altitude of -410 meters, is this correct?

This is correct. These are geopotential altitudes, see also Table 1 & 2.

p.9 l.29 Here notation "r" is used for representation ranking by Kinne et al. (2013). In some parts of the manuscript "r" is used to denote correlation coefficient so there is a conflict here. Please correct throughout the manuscript to remove the possible misunderstandings.

Yes, that is a bit unfortunate. "r" is a common symbol for correlation, which is why I use it. Kinne et al use "r" for their rankings. I will address this specifically when discussing Kinne rankings.

p.13 l.12 "G5NR and the OSSE are evaluated and found to show significant skill." This result was found for AOT, not for AAOT. Please clarify that this statement applies only to AOT to avoid misunderstandings.

I suggest to change this to: "G5NR and the OSSE are evaluated and found to show significant skill in AOT and reasonable skill in AAOT."

p.18 Figure 1 Bottom row, what are the differences between solid and dashed lines?

Caption has been clarified.

p.21 Figure 9 Please define DS. Also on the upper right corner the text is overlapping with the figure and may be difficult to read.

Caption has been clarified.

p.25 Figure 16 What are the dashed lines?
p.27 Figure 20 "r" is not defined.
p.28 Figure 21 If possible, please add the another colour bar from Fig. 7.

And thanks for the typos etc.

**Response to reviewer 4**

I'd like to thank the reviewer for their time and many useful comments. I think the paper has improved in clarity as a result of their feedback.

page 3, line 9: It would be good to add a bit more information on the simulation data, notably that it is a free running (not nudged) simulation, possibly also a word on verti- cal resolution and output frequency (hourly or even less?; how 'high-resolution' is the model data with regard to time?).

Some of this information is already in the paper but I agree that it could be stated more prominently. I will modify the text.

page 3, line 24: Replace AOD with AOT, here and throughout the manuscript; likewise for AAOD and AAOT.

Rather, I have changed AOD to AOT to preserve consistency with the many figures in the paper. I know the WMO suggests to use AOD but AOT is often used to mean the same thing. When I checked usage in publications a few years ago, AOT was actually more common than AOD. As long as I am consistent within this paper, I do not expect any confusion to arise. I hope the reviewer finds solace in the fact I have started using AOD in my most recent submissions.

page 4, line 8: What do you mean by "here we will assume a potentially remotely sensed columnar product ... and consider its representation errors"?

I agree that is an awkward sentence. What I meant was: instead of the actual surface measurement, I will assume an AERONET-like columnar measurement of AOT. The sentence has been rephrased.

page 4, line 11: Given that various definitions of "representation error" exist in the literature, it would be helpful if the author could provide the exact definition he uses in this paper (e.g. reference to another paper; formula; description).

Agreed, the references are actually in the paragraph but have been moved up.

page 4, line 14: Here it is said that this work deals mostly with yearly and some monthly averages, yet many figures show hourly data. Please clarify.

Those yearly data can be constructed from data sampled in different ways (see Table 4). The best way (in my opinion, as supported by the paper) is to resample model data to the hours of the observations and then average over a year. This was discussed in p 5, l 4-10. I will take steps to clarify this further.

page 6, line 19: What do you mean by the sing-less error? Absolute error or root- mean-square?

It is unfortunate that "absolute" can mean two different things: 1) with no reference to a baseline; 2) without a sign. Mathematically speaking: $\tau_{obs} - \tau_{area}$

(instead of $\frac{\tau_{obs} - \tau_{area}}{\tau_{area}}$)

or $|\tau_{obs} - \tau_{area}|$ . I mean the latter expression (but averaged). It is not an uncommon metric, similar to the standard deviation but it does not suffer as much from out-liers in the data.

page 7, line 4: I assume that by 'correlation' you mean R, not $R^2$. It may be helpful for the reader to explicitly say so.

I do not know how the reviewer's R is defined but I use the Pearson correlation coefficient (now explicitly mentioned in Sect 3.1),

page 8, line 22: When it is said that G5NR seems capable to realistically simulate the spatial variation of AOT and AAOT, "spatial" here seems to refer to different sites. It is not shown, it seems to me, how realistically G5NR captures the spatial variability of AOT and AAOT around a single site, including the adopted averaging distances between 0.5 and 4 degrees. It may be worthwhile to clarify this point.

I discuss this on p. 7, l. 11 but will repeat it here. The issue is of course there are no datasets available for such evaluation (DRAGON campaigns did not happen until 2012), although parts of W-Europe and the USA have several AERONET sites with distances of less than 100 km.

page 8, line 31: As grid box sizes are reduced, hourly collocation errors are reduced. Could this be because the physical connection (same cause, exchange of signal) be- tween two hourly time series at two distant points decreases with distance? Could the author comment on why the reported finding is (or is not) physically plausible?

This finding is to be expected from first principles: the comparison becomes more and more one of apples and oranges that look remarkably like apples. On the one hand, temporal sampling differences are reduced (by use of hourly protocol). On the other hand, spatial sampling differences are reduced (by decreasing box sizes).

page 9, line 5: Can something be said as to the (physical?) causes of the found east- west (North America) and north-south (Europe) gradient in representativeness?

It appears to be driven by cloudiness which, at least in the model, introduces temporal representation errors when using daily or yearly protocols.

page 9, line 21: Apart from the shorter atmospheric column, could it also matter that high lying mountain sites are often in the 'free troposphere', i.e., (somewhat) decoupled from the sources of (short lived) aerosols in the boundary layer?

This can definitely be part of the explanation for the larger representation errors for mountain sites. However, I would argue this "transport aspect" is part of the "shorter column" explanation?

page 11, line 9: Does it matter here, how missing values are treated when computing the annual mean?

For sure! When using the hourly protocol, missing data in the observational record are also removed from the G5NR data. This does not happen in the yearly protocol, resulting in large representation errors.

page 12, line 23: The author mentions once more the calculated meteorology. Overall, he seems to claim / find that meteorology is not that important for

representativeness. Is this indeed what he means to say? And, if so, how about phenomena like ENSO? Could, for example, the comparatively bad performance of South America be related to the presence / absence of ENSO in the model data?

That is not what I intend to say. Actually, meteorology is a powerful driver of both the temporal sampling of observations and the spatial distribution within an area. In previous papers (S16b and S17), I made an attempt at separating impacts of e.g. daytime/nighttime vs cloudiness and found the latter more important.

page 13, lines 13 and 20: Does this imply that meteorology is not that important for representativeness?

In line 13, I was talking about the evaluation of G5NR and not about the representation errors. In line 20, I am talking about representation errors. I believe these strong monthly correlations to be partly driven by meteorology (see also Sect 5.2 and Fig 15). However, it is difficult (maybe even impossible with the current datasets) to disentangle e.g. impacts of source distribution and wind advection. See also my answer to the previous remark by the reviewer.

page 13, line 24: It is not clear where the error of typically 20% globally comes from, I do not see this in the main text of the paper.

See e.g. Fig 5 which shows collocation errors for different boxes and protocols. For the yearly protocol, the mean sign-less error varies between 22-23%. It's important to realise that this is not a global bias: some sites will underestimate their area's average and others will over-estimate their area's average. The term "globally" has been removed and a reference to Fig 5 inserted.

Figure 1: One may add in the caption what the different line-styles in the lower row mean.

Agreed.

Figure 5: Any idea why there is an overall bias towards negative values? It seems unlikely that the (few) high lying GAW sites (and their shorter atmospheric column) alone can serve as an explanation.

Correct, negative biases arise from cloudy parts in the site's representative area: these tend to have higher AOT than the clear part (that include the site). An explanation will be added.

Figure 6: Any idea what the (physical?) reason is behind the found spatial gradients?

Cloudiness, as also explained after the reviewer's comment "page 9, line 5: Can something be said as to the (physical?) causes of the found east- west (North America) and north-south (Europe) gradient in representativeness? "

Figure 8: Any idea why Europe is so good and South America rather bad? Geography? ENSO? Number of sites? Other?

ENSO possibly. For sure a strong seasonal cycle in cloudiness that makes observations much less likely during SH autumn compared to SH winter season. This may be a quirk in the G5NR simulation, although I see something similar in the AERONET observations. Note how it is the yearly protocol (brown bar,

Fig 8) that is affected inordinately. i.e. this is driven by temporal sampling. The spatial representativeness of sites in Europe and S-America does not differ much.

Figure 12: Maybe refer in the caption to table 6 (explanation of r). Also, the figure seems to suggest that there is no connection between "r" from Kinne et al. and the relative representation error from this paper; the bars in the plot look pretty much the same for "all", "r=0", "r=1", and even "r>1" for yearly data. Please comment.

Actually, the text that refers to this Figure has more explanation. For 4 degrees, there seems to be little impact from "r", but at 1 degree higher "r"'s result in smaller representation errors. I.e. the Kinne rankings agree with my results (at least statistically). But an importand but also subtle finding is that this is only true when using the hourly protocol; Kinne et al. did not consider temporal sampling of observations in their representation rankings.

Figure 16: What are the dashed lines?

Y = 2 x  and y = x /2. Now explained in caption.

page 1, line 16: due *to* methodological choices
page 2, line 14: remove S16b
page 10, line 20: "for for" should read "for"
page 11, line 6: "Fig.??" should be properly referenced

Thanks for pointing out these typos and oversights.

[revised manuscript text omitted]
, and representation errors in this paper are relative anyway. In this paper the original, i.e. not rescaled, model data will be used it is unclear what to do about AAOT and the focus here is on relative errors anyway). Comparison with AEROCOM models shows that G5NR sulphate life-times are quite low (at 2.7 days) while the other species fairly agree with the AEROCOM multi-model mean. Clouds in G5NR show shows reasonable cloud fractions compared to CERES-SSF (Clouds and the Earth's Radiant Energy System-Single Scanner Footprints), although in the equatorial/sub-tropical region (30S-30N), G5NR has a deficit of partially cloudy scenes. In addition there are too few clouds off western continental coasts and the southern branch of the ITCZ is too strong. CALIOP (Cloud-Aerosol Lidar with Orthogonal Polarization) data suggests G5NR cloud fraction are too low, especially over equatorial/sub-tropical lands in the Northern Hemisphere, and too high in the northern polar region.

For this study, the following hourly G5NR data for 2006 were obtained: see Table 1.

**Table 1.** G5NR data used in this study

| short name | description |
| --- | --- |
| totexttau | aerosol total column extinction at 550 nm |
| totscatau | aerosol total column scattering at 550 nm |
| swtdn | TOA* downward short-wave radiation |
| cldtot | total cloud area fraction |
| phis | surface geopotential height |
| bceman | monthly anthropogenic burning BC emissions |
| bcembb | monthly biomass burning BC emissions |

*: Top Of Atmosphere

**2.2 AERONET observations & geolocations**

AERONET data were obtained from https://aeronet.gsfc.nasa.gov. For 2006, AOT from Direct Sun Version 3 L2.0 and AOT & AAOT from Inversion Version 2 L1.5 and L2.0 were logarithmically interpolated to values at 550 nm and averaged over an hour. For all years starting in 1992, geolocation data were obtained for all sites (1144 in total).

The DirectSun dataset contains only AOT (at multiple wavelengths). These observations are based on direct transmission measurements of solar light and have high accuracy of $\pm 0.01$ (Eck et al., 1999; Schmid et al., 1999). The Inversion dataset contains both AOT and AAOT (at multiple wavelengths) and these observations are based on measurements of scattered solar light from multiple directions. This inversion uses radiative transfer calculations (Dubovik and King, 2000) and yields larger errors than the DirectSun measurements. In particular, Dubovik et al. (2000) showed that Single Scattering Albedo (SSA) errors decrease with increasing AOT and estimated SSA errors of $\pm 0.03$ for water-soluble aerosol at AOT at 440 nm $\geq 0.2$ although for dust and biomass burning aerosol higher AOT at 440 nm $\geq 0.5$ were needed. Consequently, one important distinction between Inversion L1.5 and L2.0 data is a minimum threshold of AOT at 440 nm $\geq 0.4$ used in the latter (improved cloud screening is another distinction). Inversion L2.0 is a subset of the L1.5 dataset.

In the current study, only AOT at 550 nm is used and the Inversion L2.0 AOT at 440 nm criterion is adapted to AOT at 550 nm $\geq 0.25$. This is the minimum value of AOT at 550 nm present in actual Inversion L2.0 data, but also corresponds to AOT at 440 nm = 0.4 for small particles (Ångström exponent = 2.1). As a result, the OSSE in this paper is rather lenient when it comes to selecting valid observations similar to Inversion L2.0.

**2.3 GAW geolocations**

GAW geolocation data were obtained from NILU (Norwegian Institute for Air Research). Two networks were used: the GAW-AOT network which comprises 29 sun-tracking sphotometers sun photometers that measure AOT; and the GAW-ABS network which comprises 81 surface-based filter instruments . While filter instruments that measure surface properties. The real GAW-ABS network is not capable to provide of measuring a columnar (A)AOT measurements, but here we will assume a potential remotely sensed columnar product it does, similar to AERONET(A)AOT, 
[revised manuscript text omitted]

~~Representation errors for AOT do not differ much for the Direct Sun L2.0 and Inversion L1.5 products, see Fig. **??**. However, the condition of a minimal AOT (>= 0.25) for valid observations causes large but unsurprising errors for the Inversion L2.0 product. This issue with the Inversion L2.0 data is well-known but the current analysis may be the first realistic estimate of incurred errors . Figure **??** also shows results for two sensitivity studies where observational coverage in the Northern Hemisphere was artificially lowered (see discussion in last paragraph of Sect. 4) but this has no clear impact as temporal coverage is quite low anyway.~~

**Table 6.** Absolute representation errors for AERONET sites

| metric | protocol | $4^o$ | $2^o$ | $1^o$ | $0.5^o$ | $0.0625^o$ |
|--------|----------|-------|-------|-------|---------|-----------|
| mean | yearly | 0.043 | 0.042 | 0.042 | 0.042 | 0.044 |
|  | hourly | 0.021 | 0.015 | 0.011 | 0.008 | 0.000 |
| 90 % | yearly | 0.086 | 0.079 | 0.082 | 0.083 | 0.086 |
|  | hourly | 0.052 | 0.033 | 0.029 | 0.017 | 0.000 |

It is interesting to compare the representation errors of two different networks, AERONET and GAW. AERONET was not designed with representativity in mind but the GAW network was. Nevertheless, Fig. 7 suggests that GAW sites exhibit slightly larger representation errors than AERONET. In particular, GAW error statistics are strongly skewed to negative values. In the G5NR OSSE, GAW sites are located at higher altitudes and more often on isolated mountains than AERONET sites . A (G5NR site altitudes correlate very well with real altitudes, $R = 0.98$, but tend to underestimate by 28 m on average, with a standard deviation of 171 m). A look at yearly representation errors for the hourly collocation reveals a systematic altitude dependence, see Fig. 8. A high altitude site on an isolated mountain will observe a shorter atmospheric column than the surrounding grid-box which (most of which is at lower altitudes) which will cause a negative representation error, see Fig. 8. Actual site altitudes vary from -410 to 5320 m. G5NR site altitudes correlate very well ($r = 0.98$) but tend to underestimate by 28 m on average, with a random error of 171 m. Note that AERONET sites do not show this dependence on altitude for $1^o$ grid-boxes, probably because they are located more often on mountains surrounded by similar mountaineous terrain.

Previous work by Kinne et al. (2013) ranked AERONET Finally, a comparison is made with a previous study into AERONET representation errors (Kinne et al., 2013). Using a range score $r$, see Table 7, they ranked sites according to their representativity , see Table 7for larger domains. This ranking is subjective in that it is non-quantitative, based on personal knowledge of the sites and only defines representativity in broad terms. The ranking is range scores are only available for sites that had at least 5 months of data before 2008. Using the methodology of this paper, representation errors were calculated for all sites of a certain rankingrange score, see Fig. 9. For large grid boxes of $4^o$ ($\sim 450$ km near equator)), the impact of ranking the range score on representation error is quite small. While there is a visually arresting change in the error distribution for $r > 1$ (wide flanks are changed into a broader center), the mean sign-less error barely changes. This rather weak dependence on range score suggests that Kinne et al. (2013) overestimated the size of the domains ($>= 500$ km for $r > 1$) for which their sites were representative. On the other hand, for a grid-box of $1^o$

a substantial reduction in representation error can be seen for $r > 1$ sites. However, this only occurs for the hourly collocation: Kinne et al. (2013) did not consider the temporal sampling of the observations which causes large representation errors. A new An alternative ranking of representativity will be introduced in Sect. 6.

Sofar the area averages used to calculate representation errors have been derived for the entiregrid-box, both the clear and cloudy parts. Under certain circumstances, it may be more realistic to use only the clear part. Examples are the evaluation of aggregated satellite products with AERONET (like AERONET, satellites can not observed aerosol when there are clouds), or the evaluation of certain models that explicitly calculate clear-sky AOT (usually by estimating clear-sky humidity from grid-box averaged humidity). Representation errors for clear-sky parts of grid-boxes are improved for the yearly and daily collocation protocols, see Fig. 6.

**5.2 Representation errors at in monthly time-scalesAOT**

Surprisingly, monthly representation errors are not that much larger than yearly errors, see Fig.10. If monthly errors for the same site were independent and random, one would expect them to be $\sim \sqrt{12} \approx 3.5$ larger than yearly errors but that is clearly not the case. As a matter of fact, monthly errors are strongly correlated from month to month, throughout the year, see Fig. 11. The increase in correlation with January after September, is probably due to yearly cycles in meteorology and emissions and very likely to be a realistic aspect of representation errors. The implication of this is that multi-year averages may not reduce representation errors as strongly as one would hope.

This analysis also provokes the question whether representation errors (per site) should be seen as mostly biases or random errors with strong correlations (see also Schwarz et al. (2018)). Our preliminary Preliminary analysis suggests that at the monthly scale, both cases can occur. Figure ?? shows both maximum and minimum monthly errors by site as a function of yearly error.Many I.e. some sites show large variations in monthly representation errors, but significantly reduced yearly errors, suggesting that the

**Table 7.** Range scores for AERONET sites in (Kinne et al., 2013)

| range score | spatial domain | number of sites | comments |
|---|---|---|---|
| 0 | 100 km | 120 | includes mountainous sites |
| 1 | 300 km | 106 | |
| 2 | 500 km | 28 | |
| 3 | 900 km | 6 | |

 (including sign changes) in representation error from month to month, and as a consequence a strongly reduced yearly representation error. Here monthly representation errors may be interpreted as mostly random. Other sites show monthly representation errors with not much variation and as a consequence yearly representation errors are similar to the monthly errors. There the representation error is better characterised as a bias. A proper analysis of this would require significantly longer time-series of data than are currently available. Further discusion of this can be found in Sect. 6.

**5.3 Representation errors in AAOT**

 The discussion of representation errors for Inversion L1.5 AAOT  will be shorter than that for DirectSun L2.0 AOT, as the main conclusion is identical: the hourly collocation yields smaller representation errors than the other protocols, see Fig. 12. ~~As for AOT, representation errors decrease with decreasing grid-box sizes, although the decrease is small for for yearly collocation. Sgnificant positive biases can be seen for all protocols and large grid-box sizes. These biases are partly due to the AOT $> 0.25$ criterium for valid observations, which translates into an AAOT $= (1 - $SSA$)$AOT $> 0.025$ criterium for SSA $= 0.9$. However, other reasons forare the proximity of AERONET sitesto sources of absorbing aerosol and the impact of orography (e.g. see Schutgens et al. (2017) and Sect. 6. Unsurprisingly the hourly collocation protocol shows the smallest positive bias and reduces it faster for decreasing grid-box size.~~ for AAOT is more difficult to explain. It appears that a combination of conditions (location of the sites, necessity of day-light, clear skies and a minimum AOT of 0.03) together conspire to create these positive biases. Only over the Amazon can a simple explanation be found: the clear sky condition prevents many observations outside the biomass burning season, explaining large positive biases for yearly collocation (see also Fig. S5, discussed later).

 Even more than for AOT, representation errors for AAOT are very similar for the daily and hourly collocations. As discussed before, this is likely due to the absence of diurnal (anthropogenic) emissions profiles. The daily variation of AAOT is strongly underestimated by G5NR (see Sect 4 and Fig. 2).

 For completeness' sake, an analysis of AAOT representation errors for different regions (Fig. S5), different products (Fig. ~~??.The exception is for the yearly collocation protocol which allows significant biases for sites in South America and Africa.This is related to the AOT criterium for valid observations and the dominant influence of episodic biomass burning for these two continents: outside the burning season much less observations are made. Consequently the observations will favour the absorbing biomass burning aerosol.~~
[revised manuscript text omitted]

---

## Author Response (AR2)

**Reviewer 1 (Andrew Sayer)**

I'd like to thank Andy for his dedication to this paper, providing many useful comments and suggesting several improvements.

*I am posting this review under my own name (Andrew Sayer) due to ongoing collaborations with the author. I feel I am able to provide an unbiased review. I reviewed the previous version of this manuscript, and recommended major revisions (mostly organizational/clarity rather than technical points). This revised manuscript has a more readable structure and better balance of figures. My technical comments raised in the previous round of review have also been addressed. Similar suggestions were made by the other referees.*

*For me, a stand-out conclusion is that hourly collocation results in a more reasonable comparison between AERONET-like and model aerosol fields than daily or yearly. I suspect the same may be true for satellite AOD fields as well (although that is not a topic of this analysis). This suggests to me that it would be worthwhile generating hourly composites of AERONET data and satellite level 3 products, for a more direct comparison with model fields. This is important because a significant component of the representation error here is systematic due to clear sky bias (i.e. the average representation error tends to be negative, and going to hourly makes this less negative, as well as reducing the variation of the representation error). So I see this as a major action item on the data producer community.*

*A second conclusion is that going from 1 degree grid boxes to 0.5 degree grid boxes has a much smaller influence (for AOT). This suggests that the 1 degrees used by many data sets now may be a sufficient balance of detail and storage overhead. However, for AAOT it seems the difference becomes more important (mean error and mean signless error roughly halve going from 1 to 0.5 degrees). Therefore, as AAOT comparisons become more prominent, perhaps it is worth the community (modelers plus satellite data sets) making the shift from 1 to 0.5 degrees as well.*

*These two conclusions are highlighted by Figure 4 (and for North America, 5) and Table 6 for AOT, and Figure 12 for AAOT. Personally, I would suggest emphasizing this by adding to the title of the paper, "implies that hourly colocations will result in more meaningful, representative data comparisons" or something similar. This way the conclusion is even clear from the title. I leave this up to the author, however, as it's a question of style preference.*

*I do have a few further comments:*

*Figure 10: My understanding is that the left part of this Figure should match the middle part of Figure 4. Both are the representation error in yearly AOT, at a 1 degree grid, based on level 2 direct Sun AOT sampling. However, if one compares the numbers above the figure, they are not quite the same. Is this an error, or am I misunderstanding something? This should be checked and either corrected or clarified.*

Well spotted. The difference is due to the selection of stations. In Fig. 10 I included only stations that observe 12 months out of the year since I am comparing yearly to monthly errors. In Fig. 4 I included all stations, so also those that might not have any observations during some months of the year. This different selection entirely explains the small differences between Fig. 4 and 10. I have added text to the caption of Fig. 10 to explain this. Note that in both cases, only stations below 60 degree N/S latitude were used.

*Conclusions: In my previous review I had raised the question of "is a take-away that AERONET and satellites should provide hourly aggregates" and the author's response was "But don't these data already come at hourly or daily resolution?" They come at daily but not hourly, and this study suggests there can be quite some benefit from going to hourly. The Conclusions currently states that hourly colocation is beneficial, but I do think it is worthwhile to explicitly state "it would be good for data providers to provide hourly aggregates of certain parameters (e.g. AOT) to facilitate these comparisons". I doubt that many data users will be motivated to go back and create their own hourly aggregates from available highest-resolution data so if we want people to take this recommendation, I think it should be framed as a direct request to them (AERONET plus satellite). I know this study was not directly about satellites but again, if satellites are to be used as an evaluation resource for the models, the same points apply.*

I believe I was thinking of satellite data; at least for polar orbiters daily and hourly data will not be different (except at high latitudes). But the benefit of hourly (instead of daily/monthly) data is a point worth making, I will add this to the Conclusions.

*I also think it would be worth adding a couple of sentences about the possible merits of 0.5 degree grid boxes (for satellites, I think MISR and SeaWiFS are the only publicly-available data sets at this resolution?) compared to the 1 degree that is much more standard. Again, unless it is framed as an ask and data providers act on it, data users are unlikely to create them. Of course for e.g. AeroCom this is also dependent on model resolutions so perhaps that is a topic for discussion at the next AeroCom meeting. Still I would mention this in the Conclusions so it sticks in the mind of the reader.*

I am not sure my study allows conclusions to be drawn about optimal *satellite* aggregation levels. I'm assuming the truth in this case would be the 1 or 0.5 degree averaged AOD. In the case of clear-sky AOD, a perfect satellite retrieval would not show a representation error! In the case of all-sky AOD, I think it would depend on the length-scales involved: one for the obscuring cloud field; one for the observed aerosol field; which aggregation level performs better. In my 2017 ACP paper, I considered all-sky representation errors given a 1 degree satellite aggregate and truth grid box of 0.5 to 3 degrees (Fig. 15), showing that a mismatch between these sizes can easily double the representation error. But that is an answer to an altogether different question and I consider this issue of optimal satellite aggregation level unresolved.

*Other than that, I recommend that the paper be accepted for publication. I am happy to read any further revision if that would be helpful.*

**Reviewer 5 (Tero Mielonen)**

I'd like to thank Tero for his work on this paper. His thorough reading found several typing errors and inconsistencies and helped to improve this paper.

*As I'm collaborating with the author on another topic, I'm writing this review with my own name. Despite this collaboration, I believe that I'm able to provide an objective review of this manuscript.*

*In the manuscript Schutgens uses high resolution simulations (GEOS5) to evaluate the spatio-temporal representativity of AOT and AAOT observations done at AERONET and GAW sites. The topic is scientifically very interesting and the analysis is well executed. The author has satisfactorily addressed the concerns of the reviewers on the first round. I recommend that the manuscript is accepted for publication in ACP after minor revision.*

*General and specific comments:*

*Page 1, line 19: "correlate strongly throughout the year", I'm not sure if I understood this correctly. Do you mean that the monthly representation errors correlate with each other or something else?*

They correlate well with each other, i.e. January errors correlate strongly with February errors but less strongly with June errors (although the correlation is far from zero). I have modified the text.

*Introduction: Several abbreviations (e.g. AERONET, GAW, AAOT, AEROCOM) are mentioned in the text but not defined. Abbreviations should be defined in the abstract and then again at the first instance in the rest of the text.*

Done.

*Page 3, line 21: Was there a minimum number of observations required for the calculation of an hourly average? Or do you assume that even a single observation is representative enough?*

Here a single value is assumed sufficient.

*Page 3, line 37: "although for dust and biomass burning aerosol higher AOT at 440 nm ≥ 0.5 were needed", this was hard to follow. Do you mean that for dust and biomass burning aerosols the SSA errors are in the 0.03 range only for AOTs larger than 0.5?*

Yes, text has been clarified.

*Page 3, line 56: Please clarify here what are the GAW-ABS measurements and how do you calculate the AAOT from them. "Surface properties" are mentioned which makes me think about aerosol surface properties but I'm guessing you mean ground-based in-situ observations of light absorption coefficients at some wavelenght(s)? If I guessed right, then how do you calculate the AAOT from the absorption coefficients? Do you assume some kind of a vertical profile and integrate that?*

I do not use GAW observations, only their geolocations. I then assume that a future GAW-ABS network will include AERONET like stations.

*Page 4, line 47: Figure S4 doesn't seem to include any sites above 60 degrees latitude.*

Correct. The Figure stops at 60º N. I have corrected the text.

*Page 5, line 11: "the simulation captures spatial variation rather well", this seems to be true on yearly time scale but do you know if it holds for shorter time scales as well?*

Fig 1 of course suggests that temporal variation at the AERONET sites is captured reasonably well but I do not know if that also means that, say, spatial variation in monthly averages is captured well. I redid the analysis in Fig 1 for the months of January and June with the following results for mean AOT:

For a minimum of 100 observations per site:

| | Pcorr | Nr of sites |
|---|---|---|
| January | 0.89 | 47 |
| June | 0.89 | 96 |

For a minimum of 10 observations per site:

| | Pcorr | Nr of sites |
|---|---|---|

| January | 0.76 | 126 |
| June | 0.83 | 145 |

Compare this to Pcorr=0.75 and Nr=216 for a minimum of 100 observations per site in Fig. 1

I would conclude that this also holds on monthly time-scales.

*Page 5, line 16: "overestimation of dust", could you clarify here that do you mean the dust load or dust AOT? Overestimation of dust load could be explained by differences in meteorology and consequent changes in dust emissions but overestimation of dust AOT could also be influenced by the optical properties of dust used in the simulation. Did you check how the comparison looks if you separate Africa into northern and southern part? As the northern part is dominated by dust and the southern part by biomass burning aerosols, the analysis could help disentangle the contributions from dust and carbonaceous aerosols.*

This text refers to results by Gelaro et al. They studied AOT (not loads) and their analysis was based on global averages.

*Page 5, line 39: I'm not sure if you are aware, but AERONET Inversion L1.5 data has a handy flag called If_Retrieval_is_L2(without_L2_0.4_AOD_440_threshold). You could use that to relax the AOT limit but not the other requirements for L2.0.*

Good to know, I have found it in the V3 data. The flag is not present in V2 data which were used in this paper.

*Page 5, line 77: "Inversion data is generally closer to the equator", not sure what you mean with this. Do you mean that inversion data has larger SZAs even though the sites that produce the most Inversion data are close to the Equator? This could be related to the differences in the measurement principles (direct vs. almucantar).*

Yes, likely. It's a pattern in the data but I do not think it's important to the present study.

*Page 5, line 78: I didn't understand how the overestimation of dust AOT is related to the observational coverage. Could you please clarify?*

Notice that here I am talking about L2.0 Inversion data. One important constraint is that AOD@440nm > 0.4. If G5NR overestimates AOT at dust sites (which are over-represented in real Inversion data), this will lead to larger temporal coverage in the OSSE than in the real observations.

*Page 5, line 85: Instrument malfunction and maintenance will likely affect all observations, not just inversion products, so they are not likely to explain the difference between comparisons with direct and inversion data.*

True. I have adapted the text to reflect this.

*Page 6, line 25: You mentioned in the replies to the first round of reviewers that likely reason for the regional gradients is cloudiness. I believe it would be good to mention that in the text as well. This is an interesting detail because the MODIS AOT also has/had this kind of east-west trend over US. To my understanding, that was caused by land surface properties/orography.*

I think this is explained in the following paragraph, p. 6 l. 28-40

*Page 6, line 31: wet growth → hygroscopic growth*

Done.

*Page 6, line 39: Doesn't the usage of clear-sky data make sense also for the AERONET observations, especially inversion products, as they are based on observations from cloudless parts of the atmosphere?*

Yes, and this is accounted for. No simulated AERONET site can observe under cloudy conditions. But the AERONET site "sees" only a small part of the atmosphere in a 1 or 2 degree grid-box. So part of that grid-box may be cloudy and yet the site may still be able to make observations.

*Page 6, line 101: "substantial reduction in representation error can be seen for for r > 1 sites", this is true if you compare r = 0 and r > 1 sites but there doesn't seem to be such a big difference between r =1 and r > 1 sites, at leas based on the mean errors.*

> should have been >= . This has now been changed.

*Table 6: Please, clarify in the heading what does the "90 %" stand for.*

90% quantile. This has now been clarified.

*Section 5.2: There's a large gap between the heading and the text.*

Very rarely Latex will do this to make other bits and pieces fit better. I will keep an eye out for it in final production.

*Page 7, line 50: Anthropogenic emissions didn't have a diurnal cycle but biomass burning did. Did you look at the results from South America and southern Africa in this perspective? These kind of regional analysis could strengthen the conclusions here.*

An interesting idea but it would probably require substantial extra work. First there is the question of realism of the biomass burning diurnal profiles. G5NR's documentation (Putman et al. 2014) is scant on detail but I gather a very simple diurnal profile has been imposed on-line (i.e. it is not part of the QFED emission dataset). In comparison, the diurnal profiles used in Schutgens et a. 2016, 2017 were based on statistics of traffic, industry etc. and integral to the emission dataset. In these two papers, substantial differences between daily and hourly collocation were found. Second, sites affected by biomass burning are also affected by other sources and there is no easy way to separate the two contributions. At the very least one would have to identify sites affected by biomass burning. A very preliminary and coarse attempt at such an analysis yielded nothing.

That said, Fig. 2 shows that AAOT diurnal variation is under-estimated in G5NR, we know that the anthropogenic sources for absorbing aerosol have no diurnal cycle, and in Schutgens et al. 2016 (which does include realistic diurnal profiles) daily representation errors are significantly larger than hourly errors.

*Page 7, line 60: "Sect. 6, f", there's an extra "f" at the end of the line*

Corrected.

*Page 8, line 42: The shift from spatio-temporal to spatial representation errors comes rather suddenly. It would help the reader if there would be a short description (and a reference) how the spatial representation errors were calculated, either in this section or in the methods section.*

Agreed. I have included a brief explanation (at the start of the section).

*Page 8, line 69: Thank you for sharing this ranking data with easy access! Would it be possible to include also AERONET sites above 60 degrees latitude? You mentioned in the text that near the poles the simulated pixels become too small but is it an issue already at 70 or 80 degrees latitude?*

It is currently not possible to quantitatively assess when this becomes an issue but a G5NR grid-box at 60° latitude is only 50% (3.5 km) the area of a grid-box near the equator (6.9 km). A grid-box at 70° is only 34% (2.4 km) and a grid-box at 80° only 17% (1.2 km). At the same time, the real field-of-view of an AERONET site does not change and will remain in the order of 5-10 km. The current code is built on the assumption that a single G5NR grid-box represents this FOV.

*Section 6: If I understood this correctly, Kinne's ranking is based on site centered analysis whereas in this study the grid was fixed so the sites may not be in the center of the grid boxes. For finer grids this probably doesn't matter much but it might affect the results at 4 degree grid. What is your view on this?*

Correct, Kinne's original ranking is based on grid-boxes centered around the site, in the current paper, however, the grid itself is fixed and sites may be near the edges of a grid-box. Note that in the case of model or satellite L3 evaluation, people will be forced to interpret Kinne's ranking in the latter way.

The analysis presented in the current paper (Fig 9) shows the impact of r on representation errors. Unless r and the location of sites within a grid-box are in some weird and unexpected way correlated, I do not expect an impact on the conclusions that I draw.

*Page 8, line 3: You mention several examples where the behaviour of site specific representation errors differ from the "rules" defined on the basis of all sites. It would be an interesting and valuable addition if you could give some explanation why things do not go as expected. Does it depend on local meteorology, aerosol types or something else?*

Yes, yes and something else as well ☺. I do not know if G5NR provides sufficient data to disentangle these aspects but it would be a lot of work. In addition, the deeper you delve into a model, the more apparent will become its deficiencies.

While I understand the interest in the causes, I'm not sure what could be learned from it at this stage. The most important 'lesson' is that one cannot take general 'rules' derived from the entire dataset and expect them to apply to each individual site. This is not surprising: local meteorology can be expected to cause a lot of variation in representation errors even when sources remain fixed and constant.

*Page 9, line 26: "ground-based remote sensing observations", can you say it like this? If I understood correctly, the GAW absorption observations are in-situ observations.*

Yes and yes. I have assumed sun-photometers at the GAW-ABS that in reality do not exist. See also my earlier comment.

*Page 10, line 24: There's something wrong in the author list: "K??rcher"*

Corrected.

*Figures 4, 6, 7, 9, 10, 12, S2, S3, S5-S8: What are the black circle and bar? I'm guessing mean and median, but which is which?*

Bar is median, circle is mean. This is now explained at the end of Sect. 3.

*Figure 8: Did you check how the representation errors behave as a function of AOT? I think GAW stations are often designed to observe the background concentrations, meaning lower AOTs, so I'm just wondering if the difference in the altitude dependence between AERONET and GAW is caused solely by topography or do aerosol concentrations also play a role. Of course they are linked so it is hard separate their effects.*

I would expect higher GAW stations to see less AOT (shorter air column, farther from sources) and this is indeed the case. However, the correlation of representation error with AOT is lower (-0.3) than with altitude (-0.5) suggesting maybe that height is the dominant factor.

*Figure 10: The mean statistics for yearly errors in this figure are not exactly the same as in Figure 4. Shouldn't they be the same? Then another question about the monthly representation errors. How can you calculate monthly representation errors from yearly averages (the brown bar)?*

There is a small difference in selection of sites. In Fig 10 only sites that provided observations 12 months out of the year were used. This has now been explained in the caption. As to your second questions, I can't! That brown bar represents a monthly collocation protocol. Caption has been changed to reflect this.

*Figure 11: How does the number and spatial distribution of the sites change during the year? I would guess that not all sites provide data constantly throughout the year due the seasonal changes and maintenance. Would the graph look the same if all the months had the same sites?*

[revised manuscript text omitted]